



**Frazil ice growth and production during katabatic wind events in the Ross Sea, Antarctica**
Lisa De Pace[1], Madison Smith[2], Jim Thomson[2], Sharon Stammerjohn[3], Steve Ackley[4], and Brice
Loose[5]
[1]Department of Science, US Coast Guard Academy, New London CT
[2]Applied Physics Laboratory, University of Washington, Seattle WA
[3]Institute for Arctic and Alpine Research, University of Colorado at Boulder, Boulder CO
[4]University of Texas at San Antonio, San Antonio TX
[5]Graduate School of Oceanography, University of Rhode Island, Narragansett RI
*Correspondence to:* Brice Loose (bloose@uri.edu)
ABSTRACT: During katabatic wind events in the Terra Nova Bay and Ross Sea polynyas, wind
speeds exceeded 20 m s$^{-1}$, air temperatures were below -25 °C, and the mixed layer extended as
deep as 600 meters. Yet, upper ocean temperature and salinity profiles were not perfectly
homogeneous, as would be expected with vigorous convective heat loss. Instead, the profiles
revealed bulges of warm and salty water directly beneath the ocean surface and extending
downwards tens of meters. Considering both the colder air above and colder water below, we
suggest the increase in temperature and salinity reflects latent heat and salt release during
unconsolidated frazil ice production within the upper water column. We use a simplified salt
budget to analyze these anomalies to estimate in-situ frazil ice concentration between 332 x 10$^{-3}$
and 24.4 x 10$^{-3}$ kg m$^{-3}$.  Contemporaneous estimates of vertical mixing by turbulent kinetic
energy dissipation reveal rapid convection in these unstable density profiles, and mixing
lifetimes from 2 to 12 minutes. The corresponding median rate of ice production is 26 cm day$^{-1}$
and compares well with previous empirical and model estimates. Our individual estimates of ice
production up to 378 cm day$^{-1}$ reveal the intensity of short-term ice production events during the
windiest episodes of our occupation of Terra Nova Bay Polynya.



## 1. INTRODUCTION

Latent heat polynyas form in areas where prevailing winds or oceanic currents create divergence in the ice cover, leading to openings either surrounded by extensive pack ice or bounded by land on one side and pack ice on the other (coastal polynyas) (Armstrong, 1972; Park et al, 2018). The open water of polynyas is critical for air-sea heat exchange, since ice covered waters are one to two orders of magnitude better insulated (Fusco et al., 2009; Talley et al, 2011). A key feature of coastal or latent heat polynyas are katabatic winds (Figure 1), which originate as cold, dense air masses that form over the continental ice sheets of Antarctica. These air masses flow as sinking gravity currents, descending off the glaciated continent, or in the case of the Terra Nova Bay Polynya, through the Transantarctic mountain range. These flows are often funneled and strengthened by mountain-valley topography. The katabatic winds create and maintain latent heat polynyas. This research focuses on in-situ measurements taken from two coastal latent heat polynyas in the Ross Sea, the Terra Nova Bay polynya and the Ross Sea polynya.

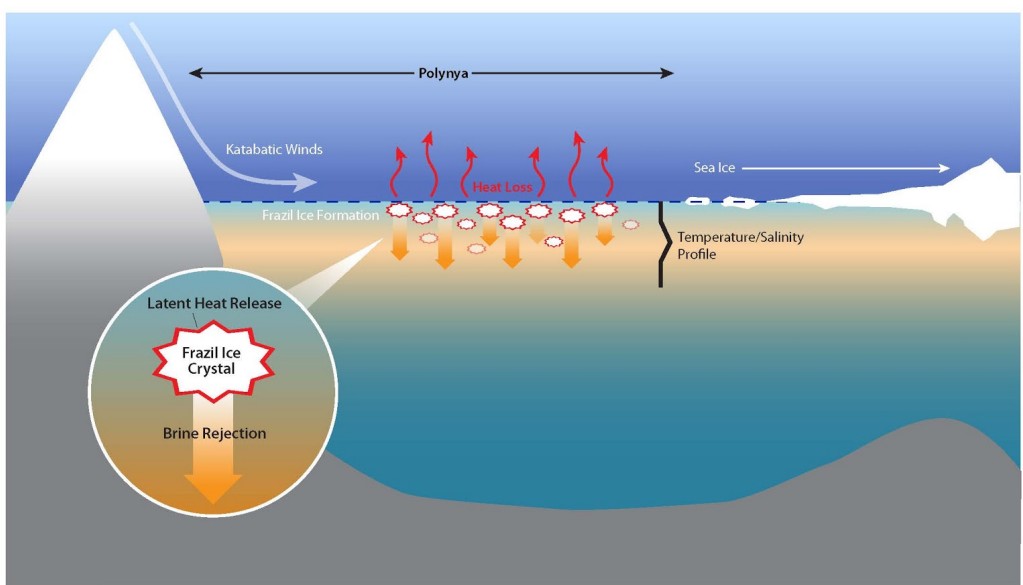

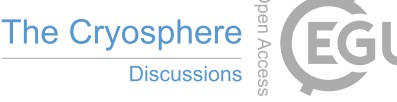

Figure 1: Schematic of a latent heat or coastal polynya. The polynya is kept open from katabic winds which drive ice advection, oceanic heat loss and frazil ice formation. Ice formation results in oceanic loss of latent heat to the atmosphere and brine rejection (Talley et al, 2011). Inset is a schematic of Frazil ice formation that depicts the release of latent heat of fusion and brine rejection as a frazil ice crystal is formed.

The extreme oceanic heat loss in polynyas can generate "supercooled" water, which is colder than the eutectic freezing point (Skogseth et al., 2009; Dmitrenk et al, 2010; Matsumura & Ohshima, 2015). Supercooled water is the precursor to ice nucleation and in-situ ice production. The first type of sea ice to appear are found as fine disc-shaped or dendritic crystals called frazil ice. These frazil ice crystals (Figure 1 inset) are about 1 to 4 millimeters in diameter and 1-100 micrometers in thickness (Heorton & Feltham, 2017; Martin, 1981; Ushio & Wakatsuchi, 1993; Wlichinsky et al., 2015). In polynyas, large net heat losses eventually lead to frazil ice production where katabic winds and cold air temperatures transport of ice crystals away from the formation site near the ocean surface and into the water column. Both conditions are achieved in polynyas by (Coachman, 1966). Katabic winds sustain the polynya by clearing frazil ice, forming pancake ice which piles up at the polynya edge to form a consolidated ice cover (Morales Maqueda et al, 2004; Ushio and Wakatsuchi, 1993).

Brine rejection (Cox & Weeks, 1983) and latent heat release during ice production, can lead to dense water formation. Over the Antarctic continental shelf, this process produces the precursor to Antarctic Bottom Water (AABW), a water mass known as High Salinity Shelf Water (HSSW) (Talley et al, 2011). In the case of the Ross Sea, the cold, dense HSSW formed on the shelf eventually becomes AABW off the shelf, the densest water in global circulation (Cosimo & Gordon, 1998; Jacobs, 2004; Martin, et al., 2007; Tamura et al.; 2007). Terra Nova Bay polynya produces especially dense HSSW, and produces approximately 1-1.5 Sv of HSSW annually (Buffoni et al., 2002; Orsi & Wiederwohl, 2009; Sansivero et al, 2017; Van Woert 1999a,b).

Given the importance of AABW to global thermohaline circulation, polynya ice production rates have been widely studied and modeled. Gallee (1997), Petrelli et al. (2008), Fusco et al. (2002), and Sansivero et al. (2017) used models to calculate polynya ice production



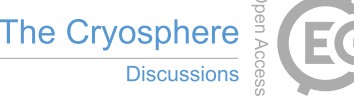

rates on the order of tens of centimeters per day. Schick (2018) and Kurtz and Bromwich (1985)
used heat fluxes to estimate polynya ice production rates, also on the order of tens of centimeters
per day.  However, quantitative estimation of polynya ice production is challenging due to the
difficulty of obtaining direct measurements (Fusco et al., 2009; Tamura et al., 2007).

**1.2 Motivation for this article**
During a late autumn oceanographic expedition to the Ross Sea as part of the PIPERS (Polynyas,
Ice Production and seasonal Evolution in the Ross Sea) project we measured CTD profiles in the
Ross Sea coastal polynyas during katabatic wind events.  Despite air temperatures that were well
below freezing and strong winds frequently in excess of the katabatic threshold, these CTD
profiles presented signatures of warmer water near the surface. The excess temperature was
accompanied by similar signatures of saltier water. During this period, we also observed long
wind rows of frazil ice. We hypothesized that the excess temperature was evidence of latent heat
of fusion from frazil ice formation, and that the excess salinity was evidence of brine rejection
from frazil ice formation.  To test these hypotheses, we had to first evaluate the fidelity of these
CTD measurements by comparing the shape and size of the profile anomalies with estimates of
the CTD precision and stability, and by using supporting evidence of the atmospheric conditions
that are thought to drive frazil ice formation (e.g. temperature and wind speed). This analysis is
described below, followed by our estimates of frazil ice concentration using the temperature and
salinity anomalies (§4). To better understand the importance of frazil formation, we computed
the lifetime of these anomalies (§5), which in turn yielded frazil ice production rates (§6).  Last,
we discuss the implications for spatial variability of ice production and application for further
polynya sea ice production estimates.


**2. STUDY AREA AND DATA**

**2.1 The Terra Nova Bay Polynya and Ross Sea Polynya**



The Ross Sea, a southern extension of the Pacific Ocean, abuts Antarctica along the
Transantarctic Mountains and has three recurring latent heat polynyas: Ross Sea polynya (RSP),
Terra Nova Bay polynya (TNBP), and McMurdo Sound polynya (MSP) (Martin et al., 2007).
The RSP is Antarctica's largest recurring polynya, the average area of the RSP is 27,000 km$^2$ but
can grow as large as  50,000 km$^2$, depending on environmental conditions (Morales Maqueda, et
al., 2004; Park et al, 2018). It is located in the central and western Ross Sea to the east of Ross
Island, adjacent to the Ross Ice Shelf (Figure 2), and typically extends the entire length of the
Ross Ice Shelf (Martin et al., 2007;  Morales Maqueda et al., 2004). TNBP is bounded to the
south by the Drygalski ice tongue, which serves to control the polynya maximum size (Petrelli et
al., 2008). TNBP and MSP,  the smallest of the three polynyas, are both located in the western
Ross Sea (Figure 2)  (Petrelli et a;., 2008). The area of TNBP, on average is 1300 km$^2$, but can
extend up to 5000 km$^2$; the oscillation period of TNBP broadening and contracting is 15-20 days
(Bromwich & Kurtz, 1984). This paper focuses primarily on TNBP and secondarily on RSP,
where our observations were taken.

During the autumn and winter season, Morales Maqueda et al., (2004) estimated TNBP
cumulative ice production to be around 40-60 meters of ice, or approximately 10% of the annual
sea ice production that occurs on the Ross Sea continental shelf.  The RSP has a lower daily ice
production rate, but produces three to six times as much as TNBP annually due to its much larger
size (Petrelli et al., 2008).



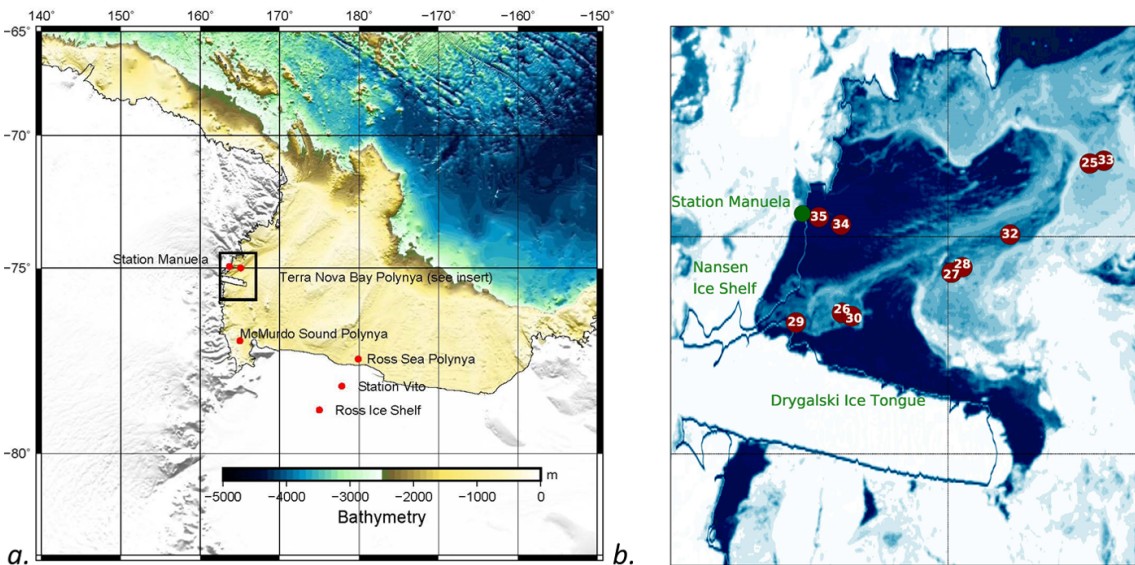

Figure 2: Map of the Ross Sea and the Terra Nova Bay Polynya. a) Overview of the Ross Sea ,
Antarctica highlighting the locations of the three recurring polynyas: Ross Sea Polynya (RSP),
Terra Nova Bay Polynya (TNBP), and McMurdo Sound Polynya (MSP). Map highlights the
2014 General Bathymetric Chart of the Oceans one-degree grid. b) Terra Nova Bay Polynya
Insert as indicated by black box in panel a. MODIS image of TNBP with the 10 CTD stations
with anomalies shown. Not included is CTD Station 40, the one station with an anomaly located
in the RSP. (CTD Station 40 is represented on Figure 2a as the location of the Ross Sea
Polynya.) Date of MODIS image is March 13, 2017; MODIS from during cruise dates could not
be used due to the lack of daylight and high cloud clover.

**2.2 PIPERS Expedition**

We collected these data during late autumn, from April 11 to June 14, 2017 aboard the

RVIB Nathaniel B. Palmer (NB Palmer, NBP17-04). More information about the research
activities during the PIPERS expedition is available at
http://www.utsa.edu/signl/pipers/index.html. Vertical profiles of Conductivity, Temperature, and
Depth (CTD) were taken at 58 stations within the Ross Sea. For the purposes of this study, we





focus on the 13 stations (CTD 23-35) that occurred within the TNBP and 4 stations (CTD 37-40)
within the RSP during katabatic wind events (Figure 2). In total, 11 of these 17 polynya stations
will be selected for use in our analysis, as described in §3.1.

**2.3 CTD measurements**

The CTD profiles were carried out using a Seabird 911 CTD (SBE 911) attached to a 24

bottle CTD rosette, which is supported and maintained by the Antarctic Support Contract (ASC).
The SBE 911 was deployed from the starboard Baltic Room. Each CTD cast contains both down
and up cast profiles. In many instances, the upcast recorded a similar thermal and haline
anomaly. However the 24 bottle CTD rosette package creates a large wake that disturbs the
readings on the upcast, so only the down cast profiles are used.

The instrument resolution is important for this study, because the anomalous profiles

were identified by comparing the near surface CTD measurements with other values within the
same profiles. The reported initial accuracy for the SBE 911 is $\pm$ 0.0003 S m$^{-1}$, $\pm$ 0.001 °C, and
0.015% of the full-scale range of pressure for conductivity, temperature, and depth respectively.
Independent of the accuracy stated above, the SBE 911 can resolve differences in conductivity,
temperature, and pressure on the order of 0.00004 S m$^{-1}$, 0.0002 °C and 0.001% of the full range,
respectively (SeaBird Scientific, 2018). The SBE 911 samples at 24 Hz with an e-folding time
response of 0.05 seconds for conductivity and temperature. The time response for pressure is
0.015 seconds.

The SBE 911 data were post-processed with post-calibrations by Seabird, following

standard protocol,  and quality control parameters. Profiles were bin-averaged at two size
intervals: one-meter depth bins and 0.1-meter depth bins, to compare whether bin averaging
influenced the heat and salt budgets. Since we observed no difference between the budget
calculations derived from one-meter vs 0.1-meter bins, the results using one-meter bins are
presented in this publication. All thermodynamic properties of seawater were evaluated via the
Gibbs Seawater toolbox, which uses the International Thermodynamic Equation Of Seawater –
2010 (TEOS-10).

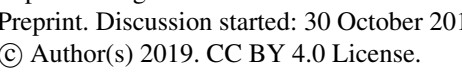

### 2.4 Weather observations

Multiple katabatic wind events were observed within the TNBP and RSP during the PIPERS expedition. Weather observations from the NB Palmer meteorological suite during these periods were compared with observations from automatic weather stations Manuela, on Inexpressible Island, and Station Vito, on the Ross Ice Shelf (Figure 2a). Observations from all three were normalized to a height of 10 meters (Figure 3). The NB Palmer was in TNB from May 1 through May 13; during this period the hourly wind speed and air temperature data from Weather Station Manuela follow the same pattern, with shipboard observations from the NB Palmer observations being lower in intensity (lower wind speed, warmer temperatures) than Station Manuela. In contrast, the wind speed and air temperature from NB Palmer during its occupation in RSP (May 16-18) is compared to Station Vito. At Station Vito, the air temperature is colder, but the wind speed is less intense. Whereas at Station Manuela (TNBP) the winds are channelized and intensified through adjacent steep mountain valleys, the winds at Station Vito (RSP) are coming off the Ross Ice Shelf, resulting in lower wind speed.

During the CTD sampling in the TNBP there were 4 periods of intense katabatic wind events, with each event lasting for at least 24 hours or longer. During the CTD sampling in the RSP there was just one event of near katabatic winds lasting about 24 hours. During each wind event, the air temperature oscillated in a similar pattern and ranged from approximately -10 °C to -30 °C.





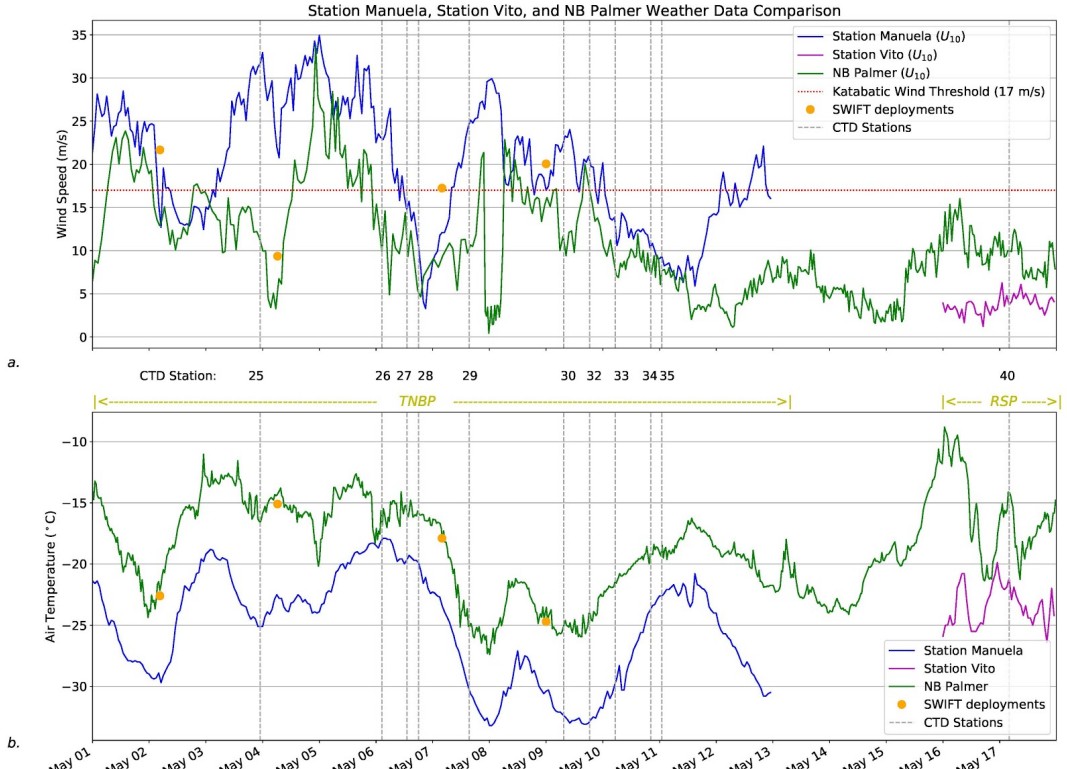

Figure 3: Weather observations from 01 May to 17 May 2017. a.) Wind speed from Station
Manuela (blue line), Station Vito (purple line), NB Palmer (green line), and SWIFT (orange
marker) deployments adjusted to 10 meters. The commonly used katabatic threshold of 17 m s$^{-1}$
is depicted as a "dotted red line", as well as the date and start time of each CTD cast. b) Air
temperature from Station Manuela, Station Vito, NB Palmer, and SWIFT deployments.


**3. EVIDENCE OF FRAZIL ICE FORMATION**

**3.1 Selection of profiles**

We used the following selection criteria to identify profiles from the two polynyas that
appeared to be influenced by frazil ice formation: (1) a deep mixed layer extending several





hundred meters (Supplemental Figure 1), (2) in-situ temperature readings below the freezing
point in the near-surface water (upper five meters), and (3) an anomalous bolus of warm and/or
salty water within the top twenty meters of the profile (Figure 4 and 5). For context, all
temperature profiles acquired during PIPERS (with the exception of one profile acquired well
north of the Ross Sea continental shelf area at 60°S, 170°E) were plotted to show how polynya
profiles compared to those outside of polynyas (Supplemental Figure 1).







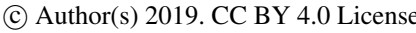



Figure 4: Conservative Temperature profiles from CTD down casts from 11 stations showing
temperature and/or salinity anomalies. Profiles (a-g) and (j-k) all show an anomalous
temperature bulge.  They also show supercooled water at the surface with the exceptions of (a)
and (j). All of the plots (a- h) have an x-axis representing a 0.02 °C change. Profiles (a-j) are
from TNBP, and (k) is from RSP.

Polynya temperature profiles were then evaluated over the top 50 meters of the water

column using criteria 2 and 3. Nine TNBP profiles and one RSP profile exhibited the excess
temperature anomalies over the top 10-20 m and near-surface temperatures close to the freezing
point (Figure 4). Excess salinity anomalies (Figure 5) were observed at the same stations with
two exceptions: Station 26 had a measurable temperature anomaly (Figure 4b) but no discernible
salinity anomaly (Figure 5b), and Station 33 had a measurable salinity anomaly (Figure 5h) but
no discernible temperature anomaly (Figure 4h). The stations of interest are listed in Table 1.

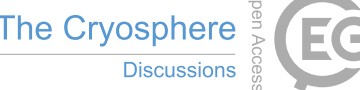





Figure 5: Absolute Salinity profiles from CTD down casts from 11 stations showing temperature
and/or salinity anomalies. Profiles (a) and (c-k) show an anomalous salinity bulge in the top
10-20 meters. Two profiles (c and g) show salinity anomalies extending below 40 meters, so the
plot was extended down to 80 meters to best highlight those. All of the plots (a-k) have an
absolute salinity range of 0.03 g kg$^{-1}$.


**3.2 Evaluating the uncertainty in the temperature and salinity anomalies**

To evaluate the uncertainty associated with the temperature and salinity anomalies at each

of the polynya stations, we compared each anomaly to the initial accuracy of the SBE 911
temperature and conductivity sensors: ± 0.001 °C and ± 0.0003 S m$^{-1}$, or ±0.00170 g kg$^{-1}$ when
converted to absolute salinity. To quantify the maximum amount of the temperature anomaly, the
baseline excursion, $\Delta T$, was calculated throughout the anomaly $\Delta T = T_{obs} - T_b$, where $T_{obs}$ is the
in-situ conservative temperature and $T_b$ is the in-situ baseline, which is extrapolated from the far
field conservative temperature within the well-mixed layer below the anomaly. Taking the single
largest baseline excursion from each of the 11 anomalous CTD profiles and averaging them, we
compute an average baseline excursion of 0.0064 ∘C. While this is a small change in the
temperature, it is still 32 times larger than the stated precision of the SBE 911 (0.0002 °C). The
same approach applied to the salinity anomalies yielded an average baseline of 0.0041 S m$^{-1}$ (or
0.0058 g kg$^{-1}$ for absolute salinity), which is 100 times larger than the instrument precision
(0.00004 S m$^{-1}$).  Table 1 lists the maximum temperature and salinity anomalies for each CTD
station.

One concern was that frazil ice crystals could interfere with the conductivity sensor. It is

conceivable that ice crystals smaller than 5 mm can be sucked into the conductivity cell, creating
spikes in the raw conductance data. Additionally, frazil crystals smaller than 100 μm are
theoretically small enough to float between the electrodes and thereby decrease the
resistance/conductance that is reported by the instrument (Skogseth & Smedsrud, 2009).  To test
for ice crystal interference, the raw (unfiltered with no bin averaging) absolute salinity profile



was plotted using raw conductivity compared with the 1-meter binned data for the 11 anomalous
CTD Stations (Supplemental Figure 2). The raw data showed varying levels of noise as well as
some spikes or excursions to lower levels of conductance; these spikes may have been due to ice
crystal interference. However, the bin-averaged data do not appear to be biased or otherwise
influenced  by the spikes, which tend to fall symmetrically around a baseline. This was
demonstrated by bin-averaging over different depth intervals as described in §2.4, Considering
the consistency of the temperature and salinity measurements within and below the anomalies,
and the repeated observation of anomalies at 11 CTD stations, we infer that the observed
anomalies are not an instrumental aberration.

**3.3  Camera observations of frazil ice formation**

During PIPERS an EISCam (Evaluative Imagery Support Camera, version 2) was

operating in time lapse mode, recording photos of the ocean surface from the bridge of the ship
every 10 minutes (for more information on the EISCam see Weissling et al, 2009). The images
from the time in TNBP and RSP reveal long streaks and large aggregations of frazil ice. A
selection of photos from TNBP were captured (Figure 6).  The winds were strong enough at all
times to generate wave fields and advect frazil ice, thus creating downstream frazil streaks, and
eventually pancake ice in most situations.  Smaller frazil streaks and a curtain of frazil ice below
the frazil streak were also visible.






a. Photo from 04- May 23:00

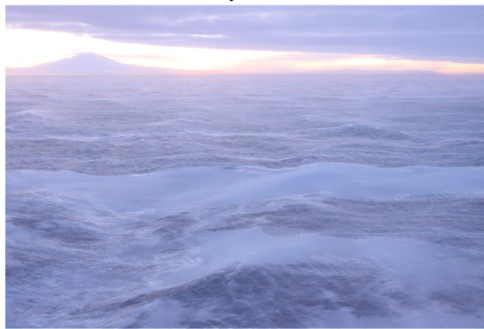

c. Photo from 05- May 01:00

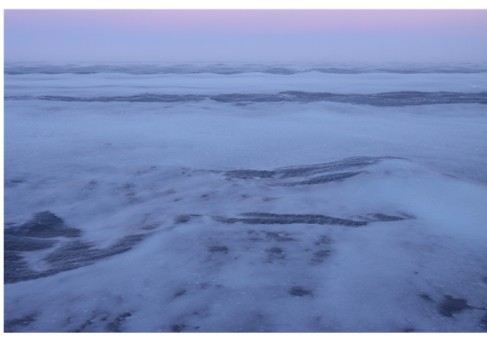

b. Photo from 05- May 02:00

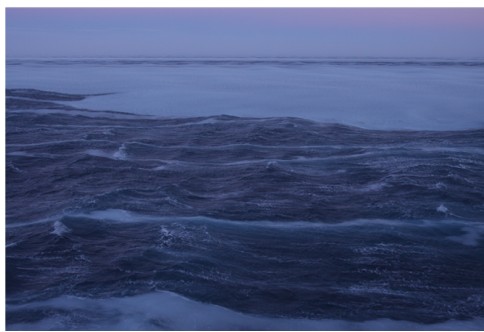

d. Photo from 06- May 22:00

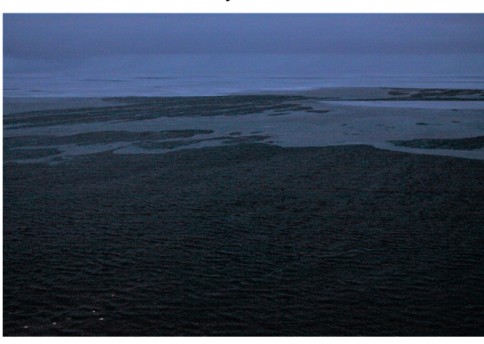

Figure 6: Images from NB Palmer as EISCam (Evaluative Imagery Support Camera) version 2.
White areas in the water are loosely consolidated frazil ice crystals being actively formed during
a katabatic wind event. Image (d) was brightened to allow for better contrast.

**3.4 Conditions for frazil ice formation during lab experiments**
Ushio and Wakatsuchi (1993) conducted laboratory experiments to reproduce the
conditions observed in polynyas. They exposed their tank, measuring 2-m length, 0.4-m width
and 0.6-m depth to air temperatures at -10 °C and wind speeds of $6\,m\,s^{-1}$ . They observed
supercooling in the range of 0.1 to 0.2 °C at the water surface and found that after 20 minutes the
rate of super-cooling slowed due to the release of latent heat, coinciding with visually observed
frazil ice formation. Simultaneously with the formation of frazil ice crystals, they observed an
increase in salinity from the brine rejection. After ten minutes of ice formation, the temperature
of the frazil ice layer was 0.07 °C warmer and the layer was 0.5 to 1.0% saltier (Ushio and
Wakatsuchi, 1993).



In this study, we found the frazil ice layer to be on average 0.0064 °C warmer than the
underlying water. Similarly, the salinity anomaly was on average 0.0058 g kg$^{-1}$ saltier, which
equates to 0.017% saltier than the water below. While our anomalies were significantly smaller
than those observed in the lab tank by Ushio and Wakatsuchi (1993), the same trend of
super-cooling, followed by frazil ice formation and the appearance of a salinity anomaly was
observed during PIPERS. However, the forcing conditions and spatial constraints of the tank
experiment likely explain why there are discrepancies between the magnitudes of the
temperature and salinity anomalies observed in the lab versus in the field.

**3.5 Temperature and salinity profiles in the presence of  platelet ice formation**
The mechanism for supercooling under ice shelves occurs via a different process than in
polynyas, but with similar impact on the water column structure. In polynyas, katabatic winds
and sub-freezing air temperatures create supercooled water near the surface, which drove frazil
ice formation. As plumes of Ice Shelf Water approached the surface, the pressure change led to
the formation of supercooled water and frazil ice formation (Jones & Wells, 2018). Robinson et
al (2017) investigated ice formation through this process under the McMurdo Sound Ice Shelf.
As the frazil crystals continue to grow, they maintained their geometry and formed platelet ice.
Robinson et al. (2017) found an increase in salinity from brine rejection and an increase in
temperature from latent heat released at the depth of ice formation. Though the mechanism for
supercooling differs, these vertical trends in temperature and salinity nonetheless are similar to
our results.

**3.6.  The anomalous profiles from TNBP and RSP appear to trace active frazil ice**
**formation**

Throughout Sections 2 and 3, we have documented that the anomalous profiles from
TNBP and RSP appear to trace frazil ice formation. In §2.4, the strong winds and sub-zero air
temperatures supported both ice formation and advection. In §3.1 and §3.2, we showed that the
CTD profiles in both temperature and salinity are reproducible and large enough to be



distinguished from the instrument noise. In §3.3 the coincident EISCam measurements reveal
significant accumulation of frazil ice crystals on the ocean surface during the time the NB
Palmer was in TNBP and RSP. In §3.4 and §3.5, we note the commonalities between the PIPERS
polynya profiles and frazil ice formation during platelet ice formation and during laboratory
experiments of frazil ice formation. Given the co-occurence of strong winds, cold air
temperatures, sub-zero water temperature, we find no simpler explanation for the apparent
warmer, saltier water near the surface in our 11 CTD profiles from TNBP and RSP. Considering
the similarity in conditions during the lab experiments and during in-situ platelet ice formation,
we conclude that our 11 profiles reflect measurable frazil ice formation in the TNBP and RSP.

## 4.0  ESTIMATION OF FRAZIL ICE CONCENTRATION USING CTD PROFILES


Having identified a collection of CTD profiles that trace frazil ice formation, we want to

know how much frazil ice formation can be inferred from these T and S profiles? Can we
attribute a large portion of polynya ice formation to this early stage of ice growth, or is the
growth of pack ice at the polynya edge the dominant process? To estimate ice formation, the
inventories of heat and salt from each profile can provide independent estimates of frazil ice
concentration. To simplify the inventory computations, we neglect the horizontal advection of
heat and salt; this is akin to assuming that lateral variations are not important because the
neighboring water parcels are also experiencing the same intense vertical gradients in heat and
salt.  We first describe the computation using temperature in § 4.1 and the computation using
salinity in § 4.2.

### 4.1 Estimation of frazil ice concentration using temperature anomalies

We used the temperature profiles to compute the "excess" heat inside the anomalies.

Utilizing the latent heat of fusion as a proxy for frazil ice production we estimated the amount of
frazil ice that must be formed in order to create observed anomalies. For each station, we first
estimated the enthalpy inside the temperature anomaly (Talley et al, 2011) as follows. Within
each CTD bin, we estimated the excess temperature as $\Delta T = T_{obs} - T_b$, where $T_{obs}$ is the in-situ



conservative temperature  and $T_b$ is the in-situ baseline or far field conservative temperature. The
excess over the baseline is graphically represented in Figure 7a.  Because we lacked multiple
profiles at the same location, we were not able to observe the time evolution of these anomalies.
Consequently, $T_b$ represents our best inference of the temperature of the water column prior to
the onset of ice formation; it is highlighted in Figure 7a with the dashed line. We established $T_b$
by looking for a near constant value of temperature in the profile directly below the temperature
bulge.  In most cases the temperature trend was nearly linear and close to the freezing point.
After selecting the starting location, the conservative temperature was averaged over 10 meters
(10 values from the 1-m binned data)  to eliminate slight variations and any selection bias.

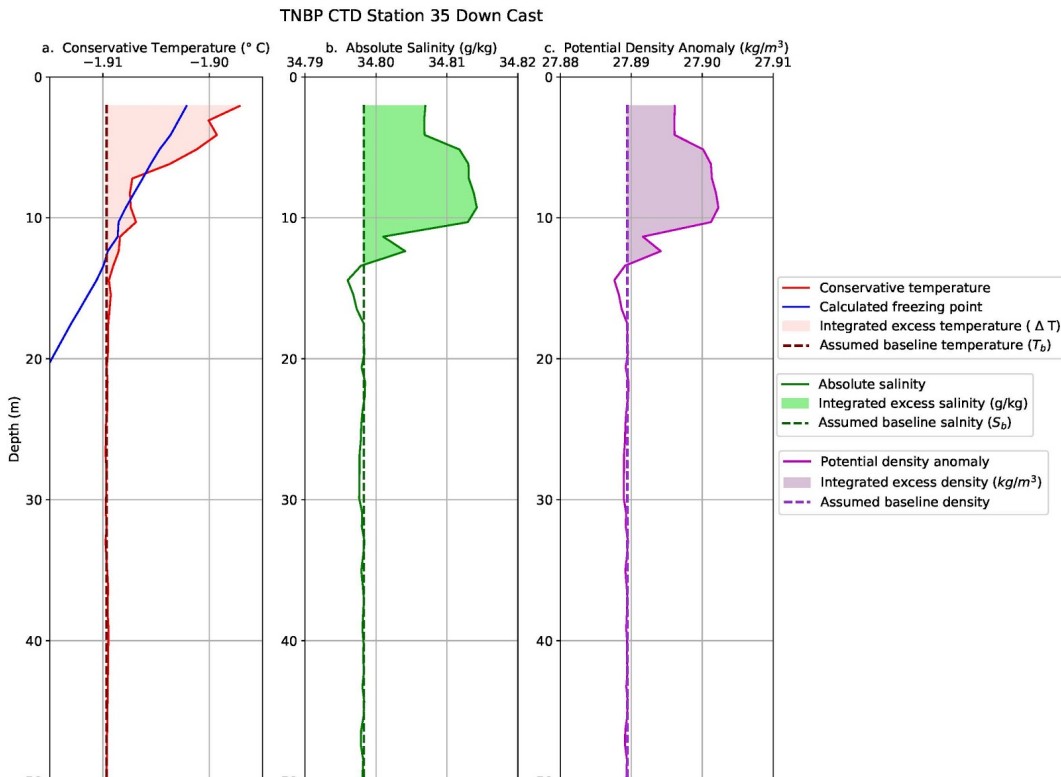

Figure 7: Conservative temperature, absolute salinity, and potential density anomaly for TNBP
CTD Station 35, May 10, 2017. a) Conservative temperature profile showing the temperature
anomaly,  the selected baseline temperature (dashed line) and the integrated excess temperature
(shaded area). b) Absolute salinity profile showing the salinity anomaly, the selected baseline



salinity (dashed line), and integrated excess salinity (shaded area). c) Potential density anomaly
showing the selected baseline density (dashed) and the excess density instability (shaded).

To find the excess heat ($Q_{excess}^{total}$) represented in the total thermal anomaly, we computed

the vertical integral of heat per unit area from the surface (z=0) to the bottom of the anomaly
(z=$z_T$):
$$Q_{excess}^{total} = \int_{z=0}^{z=z_T} \varrho \ C_p \ \Delta T \ \ dz \qquad (1)$$

Here $\varrho$ = density of seawater,   z= the depth range of the anomaly, and $C_p$ = the specific heat
capacity, The concentration of frazil ice is estimated by applying the latent heat of formation ($L_f$
=330 kJ kg$^{-1}$) as a conversion factor to $Q_{excess}^{total}$ :
$$Conc_{ice}^{temp} = \frac{Q_{excess}^{total}}{L_f * z_T} \qquad (2)$$

Where $z_T$ is the depth of the temperature anomaly in meters. The concentration of ice derived
represents the total concentration of ice, in kg m$^{-3}$. A more detailed explanation of equations 1
and 2 is contained in Supplemental 1. The mass concentration of ice derived from the
temperature anomaly for each station is listed in Table 1.

**4.2 Estimation of frazil ice concentration using salinity anomalies**

The mass of salt within the salinity anomaly was used to estimate ice formation.

Assuming that frazil ice crystals do not retain any brine and assuming there is no evaporation,
the salinity anomaly is directly proportional to the ice formed. By using the conservation of mass
equations for water and salt, the mass of frazil ice can be estimated by comparing the excess salt
(measured as salinity) with the amount of salt initially present in the profile. The conservation of
mass equations used and subsequent derivations are in Supplemental 2. The salinity anomaly
($\Delta$S) above the baseline salinity ($S_b$) is $\Delta S = S_{obs} - S_b$,  and is shown in Figure 7b. The initial
value of salinity ($S_b$) was established by observing the trend in the salinity profile directly
below the haline bulge; in most cases the salinity trend was nearly linear beneath the bulge,



however in general the salinity profiles were less homogeneous than the temperature profiles.
After selecting the starting location from below the anomaly, the absolute salinity was averaged
over the next 10 meters to establish a baseline salinity.
To find the total mass of frazil ice ($Mass_{ice}^{S}$, kg m$^{-2}$) in the water column, the integral of
each component of the salt ratio is taken over the depth range of the anomaly. This integral is
multiplied by the total mass of water per area ($Mass_{Water}^{Total}$, kg m$^{-2}$) initially in the depth range of
the anomaly. The concentration of ice ($Conc_{Ice}^{salt}$, kg m$^{-3}$) can be found by dividing the mass of
frazil ice by the depth of the salinity anomaly ($z_s$). The resulting estimates of ice concentration
are listed in Table 1.
$$Mass_{ice}^{S} = Mass_{Water}^{Total} * \frac{\int_{z=0}^{z=H} \Delta S \, dz}{\int_{z=0}^{z=H} S_{obs} \, dz} \tag{3}$$

$$Mass_{Water}^{Total} = \varrho_b * \int_{z=0}^{z=H} dz \tag{4}$$

$$Conc_{Ice}^{salt} = \frac{Mass_{Ice}^{S}}{z_S} \tag{5}$$

A more detailed explanation of equations 3, 4, and 5 is contained in Supplemental 3.

**4.3 Summary of the frazil ice estimates**

The derived ice concentrations are listed in Table 1. The inventories of salt produced
in-situ frazil ice concentrations from 24 x 10$^{-3}$ kg m$^{-3}$ to 332 x 10$^{-3}$ kg m$^{-3}$. However, it is
noteworthy that the estimates of frazil ice concentration from salt inventories are anywhere from
2 to 9 times greater than the estimates from heat inventories. The difference is likely produced by
unquantified heat heat loss to the atmosphere. The influence of sensible and long wave heat
exchanges produces an atmospheric loss term in the heat inventory, which has no corresponding
influence on the salt inventory. Therefore, we suggest that derived ice concentrations from the
heat anomalies underestimated frazil ice concentration in comparison to the salt inventory.
We also note the salinity calculation does not account for evaporation. However,
evaporation could have contributed to excess salinity while simultaneously decreasing the



temperature. Mathiot et al. (2012) found that evaporation was secondary to ice production and
contributed 4% to salt flux. In the TNBP, the Palmer meteorological tower revealed high relative
humidity (on average 78.3%), so the effects of evaporation on salinity were likely therefore
negligible. The effects of evaporation would reduce the mass of ice derived from the salinity
anomaly.

Table 1: CTD Stations with temperature and salinity anomalies (See Figures 4-5), showing
maximum values of the temperature anomaly, depth range of the temperature anomaly,
concentration of ice derived from the temperature anomaly (§4.1), as well as the maximum value
of the salinity anomaly, depth range of salinity anomaly, and concentration of ice derived from
the salinity anomaly (§4.2).

| Station | Date and Time | Maximum $\Delta T$ (°C) | $z_T$ (m) | $Conc_{ice}^T$ (kg m$^{-3}$) | Maximum $\Delta S$ (g kg$^{-1}$) | $z_S$ (m) | $Conc_{ice}^S$ (kg m$^{-3}$) |
|---|---|---|---|---|---|---|---|
| 25 | May 03 23:00:41 | 0.009 | 11.34 | 48.85 x 10$^{-3}$ | 0.004 | 13.4 | 77.76 x 10$^{-3}$ |
| 26* | May 06 02:30:08 | 0.008 | 24.73 | 16.42 x 10$^{-3}$ | -- | -- | -- |
| 27 | May 06 13:08:11 | 0.005 | 15.45 | 22.59 x 10$^{-3}$ | 0.003 | 41.22 | 48.01 x 10$^{-3}$ |
| 28 | May 06 17:59:12 | 0.007 | 15.52 | 17.85 x 10$^{-3}$ | 0.004 | 17.52 | 24.37 x 10$^{-3}$ |
| 29 | May 07 15:29:32 | 0.004 | 11.34 | 22.05 x 10$^{-3}$ | 0.007 | 21.64 | 58.55 x 10$^{-3}$ |
| 30 | May 09 07:28:24 | 0.007 | 8.24 | 24.88 x 10$^{-3}$ | 0.005 | 36.07 | 116.63 x 10$^{-3}$ |



| 32 | May 09 18:24:56 | 0.008 | 11.33 | 32.39 x 10⁻³ | 0.007 | 47.4 | 121.90 x 10⁻³ |
|---|---|---|---|---|---|---|---|
| 33** | May 10 05:16:29 | --- | --- | --- | 0.004 | 22.67 | 32.38 x 10⁻³ |
| 34 | May 10 20:16:46 | 0.004 | 13.4 | 9.63 x 10⁻³ | 0.005 | 19.58 | 80.29 x 10⁻³ |
| 35 | May 11 00:56:32 | 0.012 | 19.58 | 35.65 x 10⁻³ | 0.016 | 14.43 | 332.16 x 10⁻³ |
| 40 | May 17 04:02:37 | 0.006 | 20.61 | 34.21 x 10⁻³ | 0.003 | 18.55 | 48.84 x 10⁻³ |

*Station 26 did not have a measurable salinity anomaly but was included due to the clarity of the
temperature anomaly. Conversely, **Station 33 did not have a measurable temperature anomaly
but was included due to the clarity of the salinity anomaly.

**5.0 ESTIMATION OF TIME SCALE OF ICE PRODUCTION**

How should we interpret the lifetime of these T and S anomalies? Are they short-lived in the
absence of forcing, or do they represent an accumulation over some longer ice formation period?
One possibility is that the anomalies begin to form at the onset of the katabatic wind event,
implying that the time required to accumulate the observed heat and salt anomalies is similar to
that of a katabatic wind event (e.g. 12-48 hours). This, in turn would suggest that the estimated
frazil ice production occurred over the lifetime of the katabatic wind event. Another
interpretation is that the observed anomalies reflect the near-instantaneous production of frazil
ice. In this scenario, heat and salt are simultaneously produced and actively mixed away into the
far field. In this case, the observed temperature and salinity anomalies reflect the net difference
between production and mixing. One way to address the question of lifetime is to ask "if ice
production stopped, how long would it take for the heat and salt anomalies to dissipate?" The





answer depends on how vigorously the water column is mixing In this section, we examine the
mixing rate.  However, we can first get some indication of the timescale by  the density profiles.

**5.1  Apparent instabilities in the density profiles**

456   The computed density profiles reveal an unstable water column for all but one of our

eleven stations (Figure 8). These suggest that buoyancy production from excess heat did not
effectively offset the buoyancy loss from excess salt within each anomaly. It is not common to
directly observe water column instability without the aid of microstructure or other instruments
designed for measuring turbulence.






Figure 8: Potential density anomalies for all 11 stations with evidence of active frazil ice
formation.  The integrated excess density and assumed baseline density are depicted to highlight
the instability. Note that Station 26 (b) does not present a density anomaly because it does not
have a salinity anomaly. In the absence of excess salinity, the temperature anomaly created
instead an area of less dense water (i.e., a stable anomaly).

We suggest that an instability in the water column that persists long enough to be

measured in a CTD profile, must be the result of a continuous buoyancy loss  that is created at a
rate faster than it can be eroded by mixing. In other words, the katabatic winds appeared to
dynamically maintain these unstable profiles. Continuous ice production leads to the production
of observed heat and salt excesses at a rate that exceeds the mixing rate.  If the unstable profiles
reflect a process of continuous ice production, then the inventory of ice that we infer from our
simple heat and salt budgets must reflect ice production during a relatively short period of time,
defined by the time it would take to mix the anomalies away, once the wind-driven dynamics and
ice production stopped.

Similarly, Robinson et al (2017) found that brine rejection from platelet ice formation

(§3.5) also leads to dense water formation and a static instability. Frazil ice formation from
continually supplied ISW created a stationary instability, which was observable before being
mixed by convection to the underlying homogeneous water column that extended to 200 meters.
Similarly, the katabatic winds and cold air temperatures continually supply supercooled water to
the polynya supporting the instability.

**5.2 Lifetime of the salinity anomalies from  Monin-Obukhov length scale**

`     Turbulence theory suggests the largest eddies control the rate of turbulence dissipation
(Cushman-Rosin, 2019). A characteristic timescale, $t$, can be approximated by relating the largest
eddy size and the rate of turbulent kinetic energy dissipation ($\varepsilon$, Cushman-Rosin, 2019).
$$t \approx \frac{d}{(\varepsilon\, d)^{\frac{1}{3}}} \approx \left(\frac{d^2}{\varepsilon}\right)^{\frac{1}{3}} \tag{6}$$





Here, $d$ is the characteristic length of the largest eddy and $\varepsilon$ is the turbulent kinetic energy
dissipation rate. In this section we discuss and select the best length scale for an environment
dominated by buoyancy and wind shear. We use observed parameters to estimate the terms in
equation (6).

The dimension, $d$, of the largest eddy in a vigorously mixing water column could be

equivalent to the scale of the domain (in this case, the mixed layer depth) which was up to 600 m
in some of the PIPERS profiles (Table 2). However, a homogenous mixed-layer does not
necessarily imply active mixing throughout the layer (Lombardo and Gregg, 1989). Instead, the
characteristic length scale in an environment driven by both buoyancy and wind shear is
typically the Monin-Obukhov length ($L_{M-O}$) (Monin & Obukhov, 1954). When $L_{M-O}$ is small
and positive, buoyant forces are dominant and when $L_{M-O}$ is large and positive, wind shear
forces are dominant (Lombardo & Gregg, 1989). While the $L_{M-O}$ can be expressed using several
different estimates of shear and buoyancy, we focus on the salt-driven buoyancy flux, because
those anomalies come closest to capturing the process of frazil ice production (see §4.3 for more
detail).

$$L_{M-O} = -\frac{u_*^3}{k\beta g \overline{w\Delta S}} \qquad (7)$$

where $u_*$ is the wind-driven friction velocity at the water surface, $g$ is gravitational acceleration,
$w$ is the water vertical velocity $\overline{\Delta S}$ is the salt flux, $\beta$ is the coefficient of haline contraction, and
$k$ is the von Karman constant. A more detailed explanation, along with the specific values are
listed in Supplemental 4.

Wind-driven friction velocity is estimated using the NB Palmer wind speed ($U_{palmer}$)

record from a masthead height of $z_{palmer} = 24$ m, adjusted to a 10 meter reference ($U_{10}$) by
assuming a logarithmic profile (Manwell et al., 2010).

$$U_{10} = U_{palmer} * \frac{ln(\frac{z}{z_o})}{ln(\frac{z_{palmer}}{z_o})} \qquad (8)$$


Roughness class 0 was used in the calculation and has a roughness length of 0.0002 m. These
values are used to estimate the wind stress, $\tau$ as,
$$\tau = C_D \, \varrho_{air} U_{10}^2 \tag{9}$$

where $\varrho_{air}$ represents the density of air, with a value of 1.3406 kg m$^{-3}$ calculated using averages
from NB Palmer air temperature (-18.73 °C), air pressure (979.4 mbars) and relative humidity
(78.3%). $C_D$ represents a dimensionless drag coefficient and was calculated as $1.525 \times 10^{-3}$,
using COARE 3 code, modified to incorporate wave height and speed (Fairall et al, 2003). The
average weather data from NB Palmer was paired with the wave height and wave period from
the SWIFT deployment (defined below) on 04 May to find $C_D$. A more detailed explanation and
the specific values are listed in Supplemental 5.

We determined the aqueous friction velocity ($u_*$) at the air-sea interface using:
$$u_* = \sqrt{\frac{\tau}{\varrho_{water}}} \tag{10}$$

We used a SWIFT (Surface Wave Instrument Float with Tracking) buoy to provide
estimates of turbulent kinetic energy dissipation and vertical velocity. (Thomson et al., 2016;
Zippel & Thomson, 2016). SWIFT deployments occurred during the period of CTD
observations, as shown in the timeline of events (Supplemental Figure 3). The SWIFT
deployments do not always coincide in time and space with the CTD profiles. For the vertical
velocity estimation we identified the May 04 and May 09 SWIFT deployments as most relevant
to CTD stations analyzed here based on similarity in wind speeds. The average wind speed at all
the CTD stations with anomalies was 10.2 m s$^{-1}$. For the May 4 SWIFT deployment, the wind
speed was 9.36 m s$^{-1}$. CTD Station 32, more than two standard deviations from the average,
experienced the most intense winds of the CTD stations at 18.9 m s$^{-1}$. For CTD Station 32, the
May 9 SWIFT deployment was used, which had a wind speed of 20.05 m s$^{-1}$. For May 04 and
May 09, the average vertical velocity (w) was measured in the upper meter of the column. May
04 had an average value of $w$= 0.015 m s$^{-1}$. May 09 had an average value of $w$= 0.025 m s$^{-1}$. See





Thomson et al., 2016 & Zippel & Thomson, 2016 for details on how these measurements are
made.

The TKE dissipation rates are expected to vary with wind speed, wave height, ice

thickness and concentration (Smith & Thomson, 2019).  Wind stress ($\tau_{wind}$) is the source of
momentum to the upper ocean, but this is modulated by scaling parameter ($c_e$, Smith &
Thomson, 2019).  If the input of TKE is in balance with the TKE dissipation rate over an active
depth layer,  the following expression can be applied:

$$c_e * \tau_{wind} \propto \varrho \int \varepsilon(z)\,dz \tag{11}$$


where the density of water ($\rho$) is assumed to be 1027 kg m$^{-3}$ for all stations. The scaling
parameter incorporates both wave and ice conditions; more ice produces more efficient wind
energy transfer, while simultaneously damping surface waves, with the effective transfer velocity
in ice, based on the assumption that local wind input and dissipation are balanced (Smith &
Thompson, 2019).

$$c_e = a \left(A \frac{z_{ice}}{H_s}\right)^b \tag{12}$$
Here, A is the fractional coverage of ice, with a maximum value of 1, $z_{ice}$ is the  thickness of ice,
and $H_s$ is the significant wave height. Using Antarctic Sea ice Processes and Climate or ASPeCt
visual ice observations (www.aspect. aq) from NB Palmer, the fractional ice cover and thickness
of ice were found at the hour closest to both SWIFT deployments and CTD profiles (Knuth &
Ackley, 2006; Ozsoy-Cicek et al., 2008; Worby et al., 2008). The significant wave height for
each SWIFT deployment was used. We lacked  time series data for $H_s$ during the  time of CTD
casts, so the average value from May 04 of 0.58 m was used for all the CTD profiles. To get the
most robust data set possible, in total, 13 vertical SWIFT profiles from May 2, May 4, and May
9 were used to evaluate equation 12 over an active depth range of 0.62 meters.

Using the estimates of $c_e$, $\tau$, and $\varepsilon$ from the SWIFT, we parameterized the relationship between
wind stress and $\varepsilon$ that is reflected in equation (11). A log-linear fit ($y = 10^{(1.4572 * log10(x) +0.2299)}$,



$r^2$= 0.6554) was then applied to NB Palmer wind stress data to derive turbulent kinetic
dissipation estimates that coincided with the ambient wind conditions during each CTD station
(Table 2).

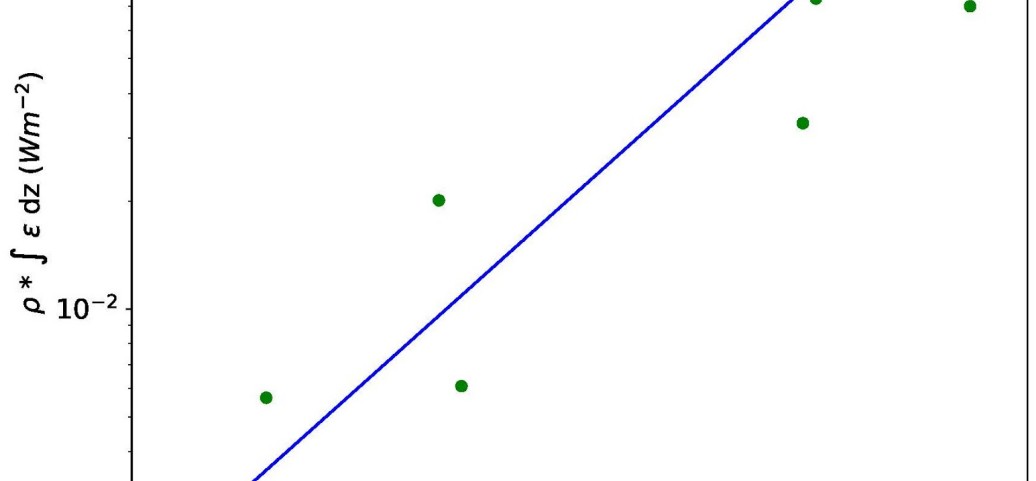

Figure 9: Logarithmic linear fit of the input flux of TKE into the ocean versus the TKE
dissipation rate over the active depth range.

Following estimation of the environmental parameters, Equation 7 can now be used to

estimate $L_{M-O}$. For these calculations a value of 0.41 was used for the von Karman constant, $k$.
Haline contraction, $\beta$, was calculated from Gibbs Seawater toolbox and averaged over the depth





range of the anomaly. The excess salt, $\overline{\Delta S}$, was found using the average value of $\Delta S$ for each
profile anomaly. The values of $L_{M-O}$ range from 6 m to 330 m (Table 2). In general, $L_{M-O}$ was
greater than the length of the salinity anomaly but smaller than the mixed layer depth. Using
$L_{M-O}$ and the estimates of $\varepsilon$, the characteristic lifetime of the salinity anomalies ranged from 2
to 12 minutes, but most values cluster near the average of 9 min. The average timescale is similar
to the frazil ice lifetime found in Michel (1967). These lifetimes suggest that frazil ice production
and the observed density instabilities relax to a neutral profile within ten minutes of a diminution
in wind forcing.

**6.0 RATE OF FRAZIL ICE PRODUCTION**

We can extend the analysis of anomaly lifetime to estimate a frazil ice production rate by
invoking the prior assumption of steady state TKE forcing and dissipation. In this case, the mass
of ice reflected by the salinity anomaly ($Conc_{ice}^{salt}$, in kg m⁻³) was produced during the time
interval corresponding to the mixing lifetime (t) that was determined from TKE dissipation in
§5.2.

$$Production\ rate\ =\ \frac{Conc_{ice}^{salt} * z_S}{t * \varrho_{ice}} \qquad (13)$$
Here, $\varrho_{ice} = 920\,\text{kg m}^{-3}$, $t$=lifetime, in days, and $z_s$ = the depth of the salinity anomaly (m).
The results are summarized in Table 2. A more detailed explanation and the specific values are
listed in Supplemental 6.

**6.1 Variability in the frazil ice production rate**

The ten estimates of frazil ice production rate, expressed as ice thickness per unit time,
ranged from 7 to 378 cm day⁻¹. These frazil ice production rates show some spatial trends across
the Terra Nova Bay polynya that correspond with variable environmental conditions in different
sectors of the polynya. As shown in Figure 10, a longitudinal gradient emerges along the axis of
the TNBP when looking at a subsection of stations under similar wind conditions Station 30



($U_{10}$=11.50 m s$^{-1}$), Station 27 ($U_{10}$=10.68 m s$^{-1}$),  and Station 25 ($U_{10}$=11.77 m s$^{-1}$).  Beginning
upstream near the Nansen Ice shelf (Station 30) and moving downstream along the  predominant
wind direction toward the northeast, the ice production rate decreases. The upstream production
rate is 69.38 cm day$^{-1}$ followed by midstream values of 28.43 cm day$^{-1}$, and lastly downstream
values of 9.83 cm day$^{-1}$. This pattern is similar to the pattern modeled by Gallee (1997). The
production rate at Station 35, was significantly higher than that at all other stations, but this large
excess is reflected in both the heat and salt anomalies.  The salt inventory at station 35  is a factor
of 2.6 greater than the nearest station (Station 34), and profiles 34 and 35 were separated in time
by less than 5 hours . This other variations in ice production rate may reflect real variability
brought on by  submesoscale fronts, eddies and other flow structures that are not easily captured
by coarse sampling.

We used the student t-distribution to derive confidence intervals for TKE dissipation rate

at each CTD station was used to bound the range of ice production rates, which are reported in
Table 2.   Uncertainty in the heat and salt inventories were not included in the uncertainty
estimates, because we observed negligible difference in the inventory while testing the inventory
for effects associated with bin averaging  bin averaging of the CTD profiles (Section 2.3).
Another small source of error arises from the neglect of evaporation. To quantify the amount of
error introduced by that assumption, we used the bulk aerodynamic formula for latent heat flux
and found the effects of evaporation across the CTD stations to be 1.8% [0.07-3.45%] (Zhang,
1997).This error due to the effects of evaporation found are similar to Mathiot et al (2012). On
average, the lower limit of ice production was  30% below the estimate and the upper limit was
some  44% larger than the estimated production.

Table 2: Summary of mass of ice derived from salinity, lifetime, and production rates.

| Station | $Conc_{ice}^{S}$ (kgm$^{-3}$) | $z_s$ (m) | $L_{M-O}$ (m) | TKE diss. $\varepsilon$ (m$^2$ s$^{-3}$) | Est MLD (m) | Lifetime (min) | Production rate (cm day$^{-1}$) | Production rate 95% CI (cm day$^{-1}$) |
|---|---|---|---|---|---|---|---|---|
| | | | | | | | | |





| 25 | 77.76 x 10$^{-3}$ | 13.4 | 140.59 | 9.648 x 10$^{-05}$ | 350 | 9.83 | 16.60 | [12.16 - 22.66] |
|---|---|---|---|---|---|---|---|---|
| 26* | -- | -- | -- | 7.191 x 10$^{-05}$ | 100 | -- | -- | -- |
| 27 | 48.01 x 10$^{-3}$ | 41.2 | 151.26 | 8.188 x 10$^{-05}$ | 500 | 10.90 | 28.43 | [20.98 - 38.51] |
| 28 | 24.37 x 10$^{-3}$ | 17.5 | 54.12 | 1.622 x 10$^{-05}$ | 600 | 9.42 | 7.09 | [4.40 - 11.45] |
| 29 | 58.55 x 10$^{-3}$ | 21.6 | 80.00 | 5.375 x 10$^{-05}$ | 275 | 8.20 | 24.19 | [17.75 - 32.96] |
| 30 | 116.63 x 10$^{-3}$ | 36 | 83.45 | 3.771 x 10$^{-05}$ | 500 | 9.49 | 69.38 | [49.34 - 97.55] |
| 32 | 121.90 x 10$^{-3}$ | 47 | 197.03 | 3.466 x 10$^{-04}$ | 375 | 8.03 | 112.57 | [68.25 -185.69] |
| 33 | 32.38 x 10$^{-3}$ | 23.7 | 98.38 | 2.844 x 10$^{-05}$ | 500 | 11.64 | 9.87 | [6.76 - 14.43] |
| 34 | 80.29 x 10$^{-3}$ | 19.6 | 65.56 | 6.397 x 10$^{-05}$ | 175 | 6.78 | 36.31 | [26.83 - 49.14] |
| 35 | 332.16 x 10$^{-3}$ | 14.4 | 6.30 | 2.343 x 10$^{-05}$ | 150 | 1.99 | 377.69 | [250.51 - 569.44] |
| 40 | 48.84 x 10$^{-3}$ | 18.6 | 174.61 | 9.603 x 10$^{-05}$ | 120 | 11.37 | 12.47 | [9.14 - 17.02] |

*Station 26 did not have a measurable salinity anomaly but was included due to the clarity of the temperature anomaly.




### 6.2 Comparison to prior model and field estimates of ice production

Calculated production rates from PIPERS ranged from 7 to 378 cm day$^{-1}$ (Figure 10). The
median ice production rate, 26.31 cm day$^{-1}$, is similar to Schick (2018), who estimated an
average ice production rate, 16.8 cm day$^{-1}$, for the month of May, (calculated using atmospheric
heat fluxes). Our median is also similar to Kurtz and Bromwich (1985), who also used a heat
budget to estimate an average ice production rate of 30 cm day$^{-1}$ for the month of May. All of
these estimates are smaller than the winter average from Sansiviero et al (2017) of 48.08 cm
day$^{-1}$ using a sea-ice model. Petrelli, Bindoff, & Bergamasco (2008) modeled a wintertime
maximum production rates of 26.4 cm day$^{-1}$ using a coupled atmospheric-sea ice model. Fusco et
al (2002) applied a model for latent heat polynyas and modeled production rate at 85 cm day$^{-1}$ for
1993 and 72 cm day$^{-1}$ for 1994.
The spatial trend we observed somewhat mimics the model 3D model of TNBP from
Gallee (1997) . During a four-day simulation, Gallee found highest ice production rates near the
coast (e.g. our Station 35) of 50 cm day$^{-1}$, and decreasing production to 0 cm day$^{-1}$ downstream
and at the outer boundaries, further west than PIPERS Station 33 (Figure 10). While some of the
ice production rates derived from PIPERS CTD profiles exceed prior results, we attribute that
excess to the relatively short time scale of these ice production "snapshots".  These estimates
integrate over minutes to tens of minutes, instead of days to months, therefore they are more
likely to capture the high frequency variability in this ephemeral process. As the katabatic winds
oscillate, the polynyas enter periods of slower ice production, driving average rates down.



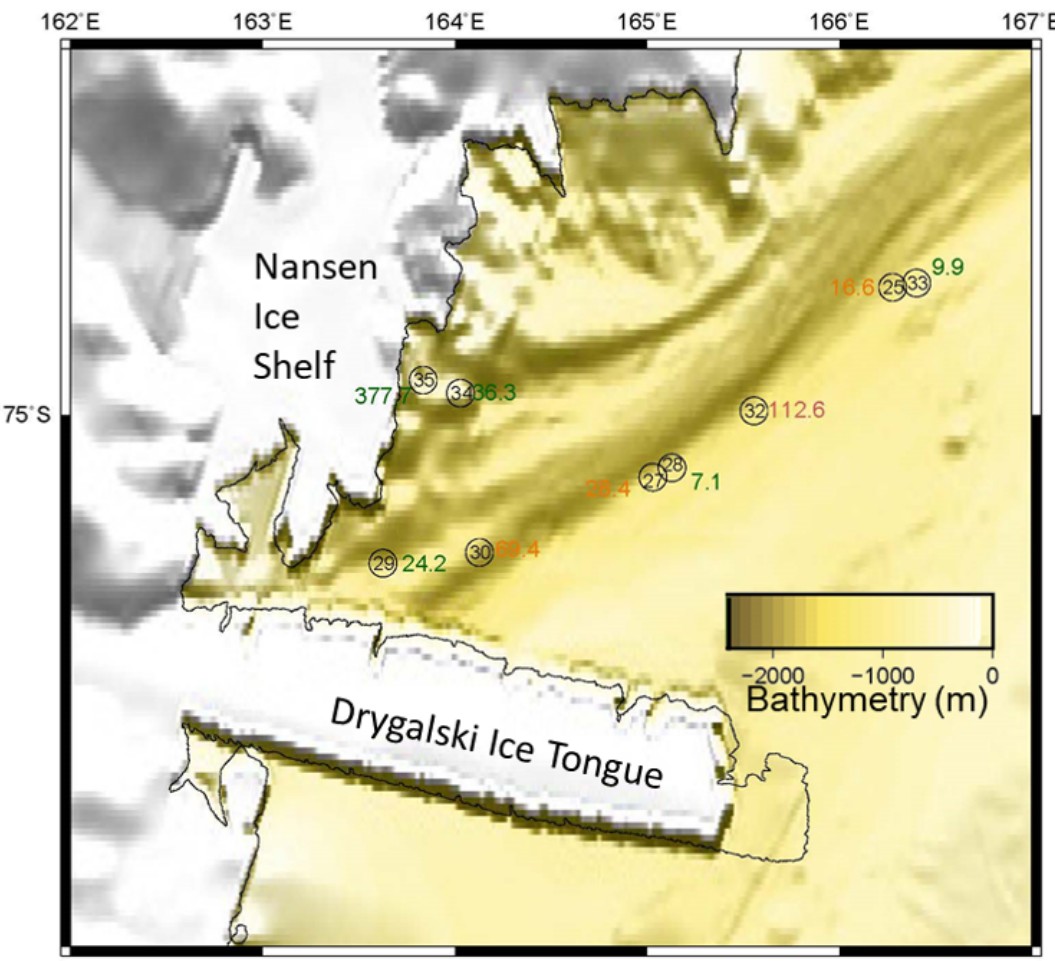

Figure 10: TNBP map of ice production rates. Map of TNBP CTD stations with anomalies and
ice production rates. The CTD station number is listed in black and circled. Listed next to the
station is the respective ice production rate in cm day$^{-1}$. The production rates are colored by wind
speed: Green indicates wind speeds less than 10 m s$^{-1}$ (Stations 28, 29, 33, 34, 35), Orange
indicates wind speeds between 10 and 15 m s$^{-1}$ (Stations 25, 27, 30), and Red indicated wind
speeds over 15 m s$^{-1}$ (Station 32).

**7. CONCLUSIONS**




Polynyas have been regarded as ice production factories with a wide range of modeled

production rates. During a late autumn oceanographic expedition to the Ross Sea, PIPERS
acquired CTD profiles in the ocean during strong katabatic wind events in both the Terra Nova
Bay polynya and the Ross Sea polynya. In those profiles we found near surface temperature and
salinity anomalies, which provided a new method for quantifying ice production rates in-situ.
Salinity and temperature anomalies observed at 11 CTD stations indicated frazil ice formation
and were used to estimate polynya ice production. Our estimated frazil ice production rates
varied from 7 to 378 cm day$^{-1}$. The wide range is likely capturing frazil ice production on very
short timescales (minutes). We note that the robustness of these estimates could be improved by
collecting consecutive CTD casts at the same location.

The polynyas in the Ross Sea show high ice production rates and are significant

contributors to Antarctic Bottom Water formation. Since 2015, sea ice extent around Antarctica
has decreased, with 2017 being an abnormally low year (Supplemental Figure 5; Fetterer et al,
2017). One of the goals of PIPERS was to understand if sea ice extent in the Ross Sea was
controlled primarily by ice production at the coast. If true, the decreased ice extent in recent
years may be related to changes in ice production in the polynyas. To further address these
questions, our estimates of polynya ice production can be paired with other ice products derived
from remote sensing, such as ice thickness from airborne and satellite lidar and ice area from
radar and passive microwave to better address the observed year-to-year changes. A decrease in
ice production rate correlates to freshening of Antarctic bottom water which would have global
impacts.









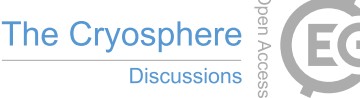

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

8. ACKNOWLEDGEMENTS

The authors appreciate the support of the National Science Foundation through NSF Award Nos
1744562 (B. Loose, L. de Pace, URI); 134717 (S.F. Ackley, UTSA); 1341513 (E. Maksym,
WHOI);1341725 (P.Guest, NPS); 1341606 (S. Stammerjohn and J. Cassano, U Colo). The
authors appreciate the support of the University of Wisconsin-Madison Automatic Weather
Station Program for the data set, data display, and information.

9. DATA AVAILABILITY

The data used in this publication are publicly available from the US Antarctic Program Data
Center  http://www.usap-dc.org/view/dataset/601192

10. AUTHOR CONTRIBUTIONS

LD prepared the manuscript including all analysis. MS and JT provided SWIFT data and
guidance for upper ocean turbulence analysis.  SS prepared and processed the PIPERS CTD data
and provided water mass insights during manuscript preparation; SA lead the PIPERS expedition

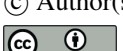



and supported ice interpretations. BL participated in PIPERS expedition, inferred possibility of
frazil ice growth and advised LD during manuscript preparation.

11. COMPETING INTERESTS

The authors declare that they have no conflict of interest."