# Peer review of "Frazil ice growth and production during katabatic wind events in the Ross Sea, Antarctica"

_The Cryosphere, 2019_

## Referee Comment (RC1) · Pat Langhorne (Referee) · 29 Nov 2019

This paper describes autumn-winter measurements within an Antarctic polynya during katabatic wind events. These data have been collected in extraordinarily unpleasant conditions and the authors are to be complemented on the number and quality of their measurements. Given the time and place and circumstances under which they were collected, such data are unique and valuable. This paper contributes to our scientific understanding of these important, but rarely observed, katabatic events, making direct observations of how ice formation takes place in these violent conditions. The authors add value by comparing their in situ measurements to those derived from other sources (model, satellite etc).

However I have a number of comments regarding the presentation of the data which I elaborate on below.

Comment 1: The notation used in the equations and particularly in the supplementary material are not consistent throughout the paper, leading to confusion. For example line 400 states that the total mass of frazil is $Mass_{ice}^{S}$. However line 81 of Supplemental states that the total mass of frazil is $Mass_{wai}^{T}$. Some work is required to please ensure consistency of definitions of symbols throughout the Supplementals and the main body of the text.

Comment 2: Is it important that the stations retain their station number from the field campaign? It would be easier for the reader to see patterns in the tables and Figure 10 if there was a simple and intuitive ordering of station numbers, say from the coast outwards.

Comment 3: Please consider the number of significant figures used in estimated values throughout the paper. For example in Tables 1 & 2 estimates are given to 4 significant figures and 2 decimal places which greatly exceeds the uncertainty in the estimate.

Comment 4: I very much appreciated the detailed laying out of calculations in the Supplementary material. However, while I followed Supplemental 1, I could not understand the derivation of $Conc_{ice}^{salt}$ in Supplemental 2 and 3. I do not understand why you use the quotient of the integrals (S3.3) to represent the integral of the quotient (i.e. the integral of (S3.2)). Please could you clarify.

Technical Corrections

line 36: I'm not sure what is meant by "one to two orders of magnitude better insulated"? Does it mean that the heat flux to the atmosphere is one to two orders of magnitude lower?

Line 54: "eutectic freezing point" ? None of the cited works use the word "eutectic". I don't know if this is strictly incorrect but I did find it confusing since the "eutectic

temperature" for sea ice is about $-36^oC$ (Vancoppenolle et al., 2019)

Vancoppenolle, M., Madec, G., Thomas, M., & McDougall, T. J. (2019). Thermodynamics of sea ice phase composition revisited. Journal of Geophysical Research:Oceans,124, 615–634. https://doi.org/10.1029/2018JC014611

Line 54: "Dmitrenko"

Lines 57-58: These are observed sizes so why not cite observations. Heorton & Feltham, 2017 and Wilchinsky et al., 2015 are modeling studies. Note Wilchinsky rather than Wlichinsky.

Line 62: incomplete sentence.

Line 64: Heorton & Feltham, 2017 and Wilchinsky et al., 2015 would fit well here. Additional relevant observational study that may be of use.

Ito, M., Ohshima, K., Fukamachi, Y., Simizu, D., Iwamoto, K., Matsumura, Y., . . . Eicken, H. (2015). Observations of supercooled water and frazil ice formation in an Arctic coastal polynya from moorings and satellite imagery. Annals of Glaciology, 56(69), 307-314. doi:10.3189/2015AoG69A839

Line 66: Suggest reference for statement re dense water formation; such as Oshima et al 2016.

Ohshima, K.I., Nihashi, S. & Iwamoto, K. Global view of sea-ice production in polynyas and its linkage to dense/bottom water formation. Geosci. Lett. 3, 13 (2016) doi:10.1186/s40562-016-0045-4

Lines 96-98: Suggest also compare with satellite observations, e.g. Oshima et al, 2016.

Line 115: Typo Petrelli et a;., 2008

Line 151: What is the implication of being deployed from the starboard Baltic Room?

[Figure]

More importantly what sort of issues arose because of sampling in supercooled waters? The very recent paper of Robinson et al (2019) may be of interest.

N.J. Robinson, B.S. Grant, C.L. Stevens, C.L. Stewart, M.J.M. Williams, Oceanographic observations in supercooled water: Protocols for mitigation of measurement errors in profiling and moored sampling, Cold Regions Science and Technology, 2019, 102954, doi.org/10.1016/j.coldregions.2019.102954.

Lines 169-171: Care needs to be taken because the magnitude of the supercooling depends on the standard used. For example Nelson et al (2017) state "in situ supercooling is larger ($\sim$ 0.003 K) when using TEOS-10 compared with EOS-80."

Nelson, M., Queste, B., Smith, I., Leonard, G., Webber, B., & Hughes, K. (2017). Measurements of Ice Shelf Water beneath the front of the Ross Ice Shelf using gliders. Annals of Glaciology, 58(74), 41-50. doi:10.1017/aog.2017.34

Line 179: How were data normalized to 10 meters? I assume log boundary layer.

Line 190: Suggest "near katabatic winds (> 10 ms$^{-1}$) lasting"

Section 3.1: The reader would have more confidence in this section is the sampling protocol was detailed (see comment on line 151).

Line 220: "plots (a-k)"

Fig 4 & Fig 5: Again more description of the temperature of the instrument when it enters the water is needed in order to interpret these figures.

Lines 248 & 250: What was the uncertainty in determining the baseline for temperature and salinity?

Line 254: Consult Nelson et al (2017) and Robinson et al (2019)

Line 258: Incorrect citation. Should be (Skogeth et al, 2009)

Lines 265-268: Check procedures with respect to Robinson et al (2019)

Line 293 & 302: remove hyphen in "super-cooling" for consistency as used "supercooling" in other places.

Line 296: "0.5 to 1 ‰ " NOT "0.5 to 1 % ". This may mean that the statement on lines 303-305 needs to be reconsidered.

Line 298-299: Consider the number of decimal places in relation to the error in the measurement.

Section 3.5: I'm not sure if this section is necessary.

Line 311: "Ice Shelf Water" is not defined. Also later in paper ISW is used and this also needs to be defined.

Line 312: "(Rees Jones & Wells, 2018)" NOT "(Jones & Wells, 2018)"

Lines 313 & 315: "Robinson et al (2014)" NOT "Robinson et al (2017)"

Line 340: Remove "?"

Line 342: "and movement"? of pack ice

Line 363-364 and 398-399: What is the "starting location"? Why 10 m? Why does 10 m eliminate selection bias? Please consider rewriting.

Equation (2): Is $Conc_{ice}^{temp}$ the same as $Conc_{ice}^{T}$ in Table 1? Please be consistent with notation.

Line 381: lower case "w"

Line 393: "Supplementals 2 and 3"

Line 400: This is an example of Comment 1 above.

Equations (3) and (4): What is H? Is this $z_S$ in the Suppplemental?

Equation (5): $Conc_{Ice}^{salt}$ the same as $Conc_{ice}^{S}$ in Table 2? Please be consistent with notation.

Lines 424-426: Surely you could argue that the humidity was high because of evaporation.

Table 1: Please see Comment 3.

Line 477: "Robinson et al (2014)" NOT "Robinson et al (2017)"

Line 479: ISW is not defined

Lines 486-487: I understood that the smallest eddies controlled the rate of dissipation. However the arguments of the energy cascade equate the rate at which energy was injected at the largest scales to the rate of energy dissipation at the smallest scales (e.g. see Fig 8.3 Cushman-Roisin, 2019). This I agree with equation (6).

Line 488: "Cushman-Roisin, 2019" NOT "Cushman-Rosin, 2019"

Line 490: Insert "TKE" after "turbulent kinetic energy"

Equation (8) & (11): I find the use of * to mean $\times$ very confusing.

Line 518: what does roughness class 0 imply? It does seem very small.

Line 534: delete "."

Lines 544, 555, 562, 563, 587: please italicize variables

Line 551: How is an "active depth layer" defined?

Line 562: insert space

Line 573: replace "A log-linear fit" with "A linear fit on a log-log scale"

Line 578: replace "A logarithmic linear fit" with "A linear fit on a log-log scale"

Lines 616-617: See Comment 3. I suggest rounding to 69, 28 and 10.

Line 621: "This other variations…"??

Line 624: Insert "CI" after "confidence interval".

Line 628: Delete" bin averaging"

Table 2: Please see Comment 3.

Table 2, column TKE diss: Why to the power "-05"? Why not just "-5"?

Table 2: Insert a note "MLD= mixed layer depth" – if it does??

Line 643: See Comment 3. I suggest rounding to 26.

Section 6.2: Note that from satellite studies Oshima et al (2016) quote an ice production rate of 8.4 m yr$^{-1}$ (from Mar-Oct) which is about 35 cm day$^{-1}$. This is close to your result.

Fig 10: This is a very interesting figure - I found it difficult to see and read the colors on top of the bathymetry color bar. I was not sure why bathymetry was needed. I wondered why it was so deep on the southern side of the Drygalski Ice Tongue? A simpler figure, an intuitive numbering of stations, and rounding of data would all make this figure have a higher impact in my opinion.

Line 721: Roisin

Line 784: D.W. Rees Jones

Line 792: Ross Sea

Line 809: Arctic

Supplementals: Please see Comments 1, 3, and 4.

I think Equation (S1.5) is meant to be in $Conc_{ice}^{temp}$

Personal dislike of use of * to mean $\times$ "times" in Supplemental 2 and 3

What is x in Table S3? I assume that $\times$ "times" is meant.
* * *

---

## Referee Comment (RC2) · Anonymous Referee #2 · 18 Mar 2020

General comments

This study has revealed extremely high ice production via underwater frazil ice formation and the importance of intense events of frazil ice production in the Antarctic polynyas, based on direct observation under PIPERS project. The finding is novel and the method/analysis are appropriate. The study demonstrates that process of underwater frazil ice formation should be properly considered in the polynya process. In addition, observed polynyas are the sites where dense water, precursor of Antarctic Bottom Water, is formed. Therefore, I have no doubt that the contents of the paper contribute to understanding of sea ice –ocean interaction and the Antarctic oceanography significantly. Therefore, I highly recommend that the paper should be published in "Cryosphere", but with moderate revision. The revising points are listed below.

[Figure]

Major comments

1. This study estimated ice production for each event and shows large variance of ice production ranging from 7 to 378 cm day-1. Although these estimates are very valuable, an important quantity is the averaged ice production or annual (monthly) ice production, which controls the formation of dense water and thus Antarctic Bottom Water. Therefore, it is desirable to infer the averaged ice production based on the two-weeks' PIPERS project. The authors took the median value of 26 cm day-1 as a representative ice production. This is better than taking the average of all the events, considering the very large variance. Even so, the median seems somewhat ad hoc way. More reasonable estimate of representative or average ice production may be possible. For example, if ice production can be related to atmospheric (and oceanic) conditions, more reasonable estimate of average ice production would be possible. Once average or monthly ice production can be inferred, then comparison or discussion with the previous satellite estimates would be possible. The present study probably suggests that the previous satellite estimation underestimated the polynya ice production.

2. Ice production has very large variance from 7 to 378 cm day-1. What are the key points (reasons) for this large variance? Brief statement for this is needed, because this seems very important part of this paper. Associated with this, as shown in Table 2, the life time is very short in the case of Stn.32. This is because Lm-o is very small. As such, the value of Lm-o has very large variance. What is the key factor for this?

Minor comments

3. Description in the paper is overall understandable. On the other hand, it is somewhat redundant and lacking in compactness. I think that the length of the paper can be reduced by 10-20%.

4. Line 362-363: Please describe the temperature trend and the starting location more specifically. Not easy to understand at this stage.

5. Line 378-379: Lf=330 kJ kg-1: What is the reference for this value? I think that use of Lf=334 kJ kg-1 is more appropriate by referring Martin (1981), which showed that frazil ice crystals do not retain any brine and thus Lf should be equal to that of freshwater. Although 330 and 334 is not so different, the basis of the value should be described in the scientific paper.

6. Line 380: Equation (2); Table 1: Conceˆtemp_ice is not understandable quantity. The total volume of frazil ice can be calculated by integration over the water column and this value can be represented by thickness of ice. This quantity is easier to understand. Heat loss occurs at the ocean surface, and thus a quantity per unit area is more meaningful than a quantity per unit volume. I know that ice production represented by thickness per day is introduced using Conceˆtemp_ice later in section 6. But integrated frazil ice thickness should be introduced at this stage. This comment is also applied to Conceˆsalt_ice (Equation 5).

7. Line 398: How did you determine the starting location from below the anomaly?

8. Line 417: Remove one of the double heat.

9. Line 489: How "t" is finally represented using the known quantities?

10. Table 2: How "the life time" is finally represented (by an equation)?

11. Figure 10: color at Stn.32 looks like purple, not red.

12. Regarding the estimate of average or annual (monthly) ice production, there have been several satellite (microwave) investigations for these polynyas (e.g., Comiso et al., 2011; Drucker et al., 2011; Nihashi and Ohshima, 2015; Tamura et al., 2016), because the satellite microwave can provide daily sea ice condition. For example, data set of monthly ice production from Nihashi and Ohshima (2015) is now in public, and can be downloaded from http://www.lowtem.hokudai.ac.jp/wwwod/polar-seaflux/. As well as comparison with the model studies as was done in section 6.2, comparison and discussion with these satellite studies would enhance the value of this paper.

References

Comiso, J. C., R. Kwok, S. Martin, and A. L. Gordon, 2011: Variability and trends in sea ice extent and ice production in the Ross Sea. J. Geophys. Res., 116, C04021, doi:10.1029/2010JC006391.

Drucker, R., S. Martin, and R. Kwok, 2011: Sea ice production and export from coastal polynyas in the Weddell and Ross Seas. Geophys. Res. Lett., 38, L17502, doi:10.1029/2011GL048668.

Nihashi, S. and K.I. Ohshima, 2015: Circumpolar mapping of Antarctic coastal polynyas and landfast sea ice: relationship and variability. Journal of Climate, 28, 3650-3670, doi:10.1175/JCLI-D-14-00369.

Tamura, T., K. I. Ohshima, A. D. Fraser and G. D. Williams, 2016: Sea ice production variability in Antarctic coastal polynyas. Journal of Geophysical Research, 121, 2967-2979, doi:10.1002/2015JC011537.
* * *

---

## Author Response (AR1)

Review 1 Comments and Responses: Comments are in black and responses are in red

This paper describes autumn-winter measurements within an Antarctic polynya during katabatic wind events. These data have been collected in extraordinarily unpleasant conditions and the authors are to be complemented on the number and quality of their measurements. Given the time and place and circumstances under which they were collected, such data are unique and valuable. This paper contributes to our scientific understanding of these important, but rarely observed, katabatic events, making direct observations of how ice formation takes place in these violent conditions. The authors add value by comparing their in situ measurements to those derived from other sources (model, satellite etc).

However I have a number of comments regarding the presentation of the data which I elaborate on below.

Comment 1: The notation used in the equations and particularly in the supplementary material are not consistent throughout the paper, leading to confusion. For example line 400 states that the total mass of frazil is MassSice. However line 81 of Supplemental states that the total mass of frazil is MassTwai. Some work is required to please ensure consistency of definitions of symbols throughout the Supplementals and the main body of the text.

Thank you, all equations have been fixed and reviewed for internal consistency.

Comment 2: Is it important that the stations retain their station number from the field campaign? It would be easier for the reader to see patterns in the tables and Figure 10 if there was a simple and intuitive ordering of station numbers, say from the coast outwards.

We recognize that a sequential numbering system for the stations would be more logical. However, we also think there is a lot of value in being able to relate these data back to the hydrographic data that is stored in the public domain. For this reason, we argue that it is worthwhile to retain the original numbering so that they would match the station numbers in the public repository.

We have included a sentence in Section 2.2 to explain the enumeration: "CTD station numbers follow the original enumeration used during NBP17-04, so they are more easily traceable to the hydrographic data, which is archived as described below in the Data Availability section."

Comment 3: Please consider the number of significant figures used in estimated values throughout the paper. For example in Tables 1 & 2 estimates are given to 4 significant figures and 2 decimal places which greatly exceeds the uncertainty in the estimate.
Thank you, we have corrected the number of significant figures used throughout the tables.

Comment 4: I very much appreciated the detailed laying out of calculations in the Supplementary material. However, while I followed Supplemental 1, I could not understand the derivation of Concsalt in Supplemental 2 and 3. I do not understand why you use ice the quotient of the integrals (S3.3) to represent the integral of the quotient (i.e. the integral of (S3.2)). Please could you clarify.

Thank you for catching the error on our derivation of the frazil mass from the salinity anomaly. We agree with your assessment that we had applied the integral incorrectly when going from Step S3.2 to Step S3.3. Supplemental 3 has been changed to correct the formula. All calculations were redone and code was double checked.

The correction led to minor changes in the mass of ice and the concentration of ice, but those changes in the bulk inventories were not large enough to alter our interpretations.

Technical Corrections
line 36: I'm not sure what is meant by "one to two orders of magnitude better insulated"? Does it mean that the heat flux to the atmosphere is one to two orders of magnitude lower?
Thank you, edited for clarity.
Line 54: "eutectic freezing point" ? None of the cited works use the word "eutectic". I don't know if this is strictly incorrect but I did find it confusing since the "eutectic temperature" for sea ice is about −36oC (Vancoppenolle et al., 2019)
Vancoppenolle, M., Madec, G., Thomas, M., & McDougall, T. J. (2019). Thermodynamics of sea ice phase composition revisited. Journal of Geophysical Re- search:Oceans,124, 615–634. https://doi.org/10.1029/2018JC014611
Thank you, edited for clarity and removed.
Line 54: "Dmitrenko"
Thank you, corrected.
Lines 57-58: These are observed sizes so why not cite observations. Heorton & Feltham, 2017 and Wilchinsky et al., 2015 are modeling studies. Note Wilchinsky rather than Wlichinsky.
Thank you, corrected.
Line 62: incomplete sentence.
Thank you, corrected.
Line 64: Heorton & Feltham, 2017 and Wilchinsky et al., 2015 would fit well here. Additional relevant observational study that may be of use.
Ito, M., Ohshima, K., Fukamachi, Y., Simizu, D., Iwamoto, K., Matsumura, Y., . . . Eicken, H. (2015). Observations of supercooled water and frazil ice formation in an Arc- tic coastal polynya from moorings and satellite imagery. Annals of Glaciology, 56(69), 307-314. doi:10.3189/2015AoG69A839
Thank you, corrected and added.
Line 66: Suggest reference for statement re dense water formation; such as Oshima et al 2016.
Ohshima, K.I., Nihashi, S. & Iwamoto, K. Global view of sea-ice production in polynyas and its linkage to dense/bottom water formation. Geosci. Lett. 3, 13 (2016) doi:10.1186/s40562-016-0045-4
Thank you, corrected and added.
Lines 96-98: Suggest also compare with satellite observations, e.g. Oshima et al, 2016.
We added this paper and a few other satellite observation papers.

We have heavily revised section 6.2 – the discussion of previous sea ice production estimates. That section includes this paragraph on remote sensing: "Overall, these ice production estimates from in-situ data are larger than the seasonal production estimates derived from remote sensing products. Drucker et al (2011) used the AMSR-E instrument to obtain a seasonal average of 12 cm day$^{-1}$ for years 2003-2008. Oshima et al, (2016) estimated 6 cm day$^{-1}$ of seasonal production for the years 2003-2011, and Nihashi and Ohshima (2015) determined 7 cm day$^{-1}$ for years 2003-2010. Finally, Tamura et al (2016) found production rates that ranged from 7-13 cm day$^{-1}$, using both ECMWF and NCEP Reanalysis products for 1992-2013, reflecting a greater degree of consistency in successive estimates, likely because of consistency in the estimation methods. "

Thank you for pointing us to the paper. We have added the comparison to microwave sensing production rates.

Line 115: Typo Petrelli et a;., 2008
Thank you, corrected.
**Line 151: What is the implication of being deployed from the starboard Baltic Room? More importantly what sort of issues arose because of sampling in supercooled wa- ters? The very recent paper of Robinson et al (2019) may be of interest.**

**Thank you. This was a great paper to review and has been added to our references.** The paper outlines two potential sources of bias that are a concern for us that are explained in detail here and have been added to Section 3.2:

1. Self-heating where the thermistor reads warmer than the water because of the heat that remains in the housing, etc. We did keep the CTD rosette at room temp, so there is a risk of this.
2. Ice formation on surfaces in the conductivity cell. We don't see this as a risk because of (1) - the sensor was warm before it went over the side.
3. In the first draft, we examined and discussed the potential for frazil ice crystals passing thru the conductivity cell.

We think (2) did not take place because the cell was filled with saltwater prior to deployment. Freezing did take place at the beginning of the expedition, but this can be very damaging to a conductivity cell so steps were taken to avoid it.

Additionally, conductivity/salinity was increasing in our profiles. This is opposite the trend that Robinson et al (2019) observed. We address this question in more detail within section 3.2.

The protocol was to complete 2-3 minutes of soak time at around 10 m, until the spikes in the conductivity cell have completely gone away. We believe this dissipates much of the thermal inertia, although a 10 minute soak time would have been better, the results suggest that 2-3 minutes will dissipate 70-80% of that excess heat in the sensor body.

While Robinson recommended using upcasts to avoid the thermal problem, this is complicated because the CTD sits at the bottom of the rosette, so the upcast can be influenced by turbulence and smoothing around the large 24 bottle, 2 ton rosette package.

Nevertheless, we examined upcasts and found that many of them were consistent with the results form the downcast – both for temperature and salinity.  We have included a figure in the supplemental that highlights this.

The below figure was added as Supplemental Figure 1. For Station 25 you can see the salinity and temperature anomaly is reduced for the up cast which we attribute to the wake. For station 32, there is missing data in the up cast which would have impacted our calculations.

[Figure]

Lines 169-171: Care needs to be taken because the magnitude of the supercooling depends on the standard used. For example Nelson et al (2017) state "in situ super- cooling is larger (~ 0.003 K) when using TEOS-10 compared with EOS-80."

Nelson, M., Queste, B., Smith, I., Leonard, G., Webber, B., & Hughes, K. (2017). Measurements of Ice Shelf Water beneath the front of the Ross Ice Shelf using gliders. Annals of Glaciology, 58(74), 41-50. doi:10.1017/aog.2017.34

We have added a sentence in Section 2.3 stating that the choice of empirical relationship can affect the absolute freezing point calculation and we have included this citation, thank you for pointing this out.

Line 179: How were data normalized to 10 meters? I assume log boundary layer.
Thank you, correct we used a logarithmic wind profile.

Line 190: Suggest "near katabatic winds (> 10 ms−1) lasting"
Thank you, corrected.
Section 3.1: The reader would have more confidence in this section is the sampling protocol was detailed (see comment on line 151).
Thank you. We have added more details to our sampling procedures.
Line 220: "plots (a-k)"
Thank you , corrected.
Fig 4 & Fig 5: Again more description of the temperature of the instrument when it enters the water is needed in order to interpret these figures.
Lines 248 & 250: What was the uncertainty in determining the baseline for temperature and salinity?
Line 254: Consult Nelson et al (2017) and Robinson et al (2019)

Please refer to our related responses above and in the revised manuscript.

We have revised Section 2.3 to be more descriptive with the CTD sampling procedure.

Line 258: Incorrect citation. Should be (Skogeth et al, 2009)
Thank you, corrected.

Lines 265-268: Check procedures with respect to Robinson et al (2019)
We did not find any reference to or guidance on averaging procedures Robinson et al (2020).  As discussed, we investigated the effect of averaging over different vertical intervals and found no systematic influence.

Line 293 & 302: remove hyphen in "super-cooling" for consistency as used "supercool- ing" in other places.
Thank you, corrected
Line 296: "0.5 to 1 ‰ " NOT "0.5 to 1 % ". This may mean that the statement on lines 303-305 needs to be reconsidered.
Thank you, corrected. I updated it to use g kg-1 to be consistent throughout the paper.
Line 298-299: Consider the number of decimal places in relation to the error in the measurement.
Thank you both reduced to the appropriate number of significant figures.
Section 3.5: I'm not sure if this section is necessary.
Thank you eliminated in favor of shortening the article.
Line 311: "Ice Shelf Water" is not defined. Also later in paper ISW is used and this also needs to be defined.
Thank you, defined. Section Removed.
Line 312: "(Rees Jones & Wells, 2018)" NOT "(Jones & Wells, 2018)"

Thank you. Section Removed.

Lines 313 & 315: "Robinson et al (2014)" NOT "Robinson et al (2017)"

Thank you. Section Removed.

Line 340: Remove "?"

Thank you corrected.

Line 342: "and movement"? of pack ice

Thank you, sentence removed in favor of shortening the article.

Line 363-364 and 398-399: What is the "starting location"? Why 10 m? Why does 10 m eliminate selection bias? Please consider rewriting.

Thank you, reworded. The variance in the temperature and salinity was less than the order the precision of the instrument.    We cited the precision in Section

Equation (2): Is Conctemp the same as ConcT in Table 1? Please be consistent with ice ice notation.

Thank you corrected.

Line 381: lower case "w"

Thank you corrected.

Line 393: "Supplementals 2 and 3"

Thank you corrected.

Line 400: This is an example of Comment 1 above.

Thank you corrected.

Equations (3) and (4): What is H? Is this zS in the Suppplemental?

Thank you corrected to reflect the integral from the surface to zS. H was removed for clarity and consistency.

Equation (5): Concsalt the same as ConcS in Table 2? Please be consistent with Ice ice notation.

Thank you corrected.

Lines 424-426: Surely you could argue that the humidity was high because of evapo- ration.

Thank you edited for clarity.

Table 1: Please see Comment 3.

Line 477: "Robinson et al (2014)" NOT "Robinson et al (2017)"

Thank you corrected.

Line 479: ISW is not defined

Thank you corrected and defined.

Lines 486-487: I understood that the smallest eddies controlled the rate of dissipation. However the arguments of the energy cascade equate the rate at which energy was injected at the largest scales to the rate of energy dissipation at the smallest scales (e.g. see Fig 8.3 Cushman-Roisin, 2019). This I agree with equation (6).

Thank you. Clarified and corrected.

Line 488: "Cushman-Roisin, 2019" NOT "Cushman-Rosin, 2019"

Thank you corrected.

Line 490: Insert "TKE" after "turbulent kinetic energy"

Thank you corrected.

Equation (8) & (11): I find the use of * to mean × very confusing.

Thank you. It has been removed from all equations.
Line 518: what does roughness class 0 imply? It does seem very small.
Roughness class 0 implies a ocean or sea surface
Khalfa, Dalila & Abdelouahab, Benretem & Herous, Lazhar & Issam, Meghlaoui. (2014).
Evaluation of the adequacy of the wind speed extrapolation laws for two different roughness
meteorological sites. American Journal of Applied Sciences. 11570583. 570-583.
10.3844/ajassp.2014.570.583

Line 534: delete ".".
Thank you corrected.
Lines 544, 555, 562, 563, 587: please italicize variables
Thank you corrected.

Line 551: How is an "active depth layer" defined?
Thank you, edited for clarification.
Line 562: insert space
Thank you corrected.
Line 573: replace "A log-linear fit" with "A linear fit on a log-log scale"
Thank you corrected.
Line 578: replace "A logarithmic linear fit" with "A linear fit on a log-log scale"
Thank you corrected.
Lines 616-617: See Comment 3. I suggest rounding to 69, 28 and 10.

Line 621: "This other variations. . ."??
We have revised the wording.

Line 624: Insert "CI" after "confidence interval".
Thank you corrected.

Line 628: Delete" bin averaging"
Thank you corrected.

Table 2: Please see Comment 3.
Table 2, column TKE diss: Why to the power "-05"? Why not just "-5"?
Thank you corrected.
Table 2: Insert a note "MLD= mixed layer depth" – if it does??
It does, Thank you added.

Line 643: See Comment 3. I suggest rounding to 26.
Section 6.2: Note that from satellite studies Oshima et al (2016) quote an ice production rate of
8.4 m yr−1 (from Mar-Oct) which is about 35 cm day−1. This is close to your result.
Fig 10: This is a very interesting figure - I found it difficult to see and read the colors on top of
the bathymetry color bar. I was not sure why bathymetry was needed. I wondered why it was so
deep on the southern side of the Drygalski Ice Tongue? A simpler figure, an intuitive numbering
of stations, and rounding of data would all make this figure have a higher impact in my opinion.

Thank you the figure was modified to make it easier to read the sea ice production rates.

We have revised this section (6.2) significantly to discuss the ice production rates.

Line 721: Roisin
Thank you corrected.

Line 784: D.W. Rees Jones
Thank you, reference removed since the context was removed.
Line 792: Ross Sea Thank you corrected.

Line 809: Arctic Thank you corrected.

Supplementals: Please see Comments 1, 3, and 4.
I think Equation (S1.5) is meant to be in Conctemp ice Thank you corrected.

Personal dislike of use of * to mean × "times" in Supplemental 2 and 3. Thank you, all removed.
 What is x in Table S3? I assume that × "times" is meant. Yes, italics removed and spaces added.
Thank you

Review 2 Comments and Responses: Comments are in black and responses are in red

General comments
This study has revealed extremely high ice production via underwater frazil ice formation and the importance of intense events of frazil ice production in the Antarctic polynyas, based on direct observation under PIPERS project. The finding is novel and the method/analysis are appropriate. The study demonstrates that process of under- water frazil ice formation should be properly considered in the polynya process. In addition, observed polynyas are the sites where dense water, precursor of Antarctic Bottom Water, is formed. Therefore, I have no doubt that the contents of the paper contribute to understanding of sea ice –ocean interaction and the Antarctic oceanography significantly. Therefore, I highly recommend that the paper should be published in "Cryosphere", but with moderate revision. The revising points are listed below.

Major comments
1. This study estimated ice production for each event and shows large variance of ice production ranging from 7 to 378 cm day-1. Although these estimates are very valuable, an important quantity is the averaged ice production or annual (monthly) ice production, which controls the formation of dense water and thus Antarctic Bottom Water. Therefore, it is desirable to infer the averaged ice production based on the two-weeks' PIPERS project. The authors took the median value of 26 cm day-1 as a representative ice production. This is better than taking the average of all the events, considering the very large variance. Even so, the median seems somewhat ad hoc way. More reasonable estimate of representative or average ice production may be possible. For example, if ice production can be related to atmospheric (and oceanic) conditions, more reasonable estimate of average ice production would be possible. Once average or monthly ice production can be inferred, then comparison or discussion with the previous satellite estimates would be possible. The present study probably suggests that the previous satellite estimation underestimated the polynya ice production.

Thank you for the comment. We found that the production rate varied with respect to the wind and with respect to the location in the polynya. There was a direct relationship between wind speed and production rate. There was an inverse relationship between the distance from the coastline and the production rate.

We have taken careful consideration to produce the requested up-scaling to a seasonal average. This includes neglecting Station 35 as an outlier, because of possible ice shelf influence (See Section 6.0 for discussion). We have added a new section to the discussion, titled " 6.1 Seasonal Ice Production", which describes the method for up-scaling. Additional detail on the computation of the seasonal average can be found in Supplemental 7 and Supplemental Figure 6.

The results yielded a seasonal ice production of 29 cm day$^{-1}$.

2. Ice production has very large variance from 7 to 378 cm day-1. What are the key points (reasons) for this large variance? Brief statement for this is needed, because this seems very important part of this paper. Associated with this, as shown in Table 2, the life time is very short in the case of Stn.32. This is because Lm-o is very small. As such, the value of Lm-o has very large variance. What is the key factor for this?

The large variance is due in part to varying wind conditions and varied geographic position. The large difference at station 32 is due to a difference in the turbulent kinetic energy dissipation rate. It varies from the other stations by one order of magnitude. Station 32 experiences the most wind stress and a different SWIFT deployment was used to derive the TKE dissipation.
For station 35, the LMO is very small. That is due to a higher salt flux at that station and slower wind speeds. Station 35 represented the highest salt flux and the second smallest wind speed/stress. When the LMO is small mixing is buoyancy dominated, as opposed to wind shear dominated. We feel that the buoyancy is likely dominant due to ISW contributions.

Minor comments
3. Description in the paper is overall understandable. On the other hand, it is somewhat redundant and lacking in compactness. I think that the length of the paper can be reduced by 10-20%.
4. Line 362-363: Please describe the temperature trend and the starting location more specifically. Not easy to understand at this stage.
Thank you clarified.
5. Line 378-379: Lf=330 kJ kg-1: What is the reference for this value? I think that use of Lf=334 kJ kg-1 is more appropriate by referring Martin (1981), which showed that frazil ice crystals do not retain any brine and thus Lf should be equal to that of freshwater. Although 330 and 334 is not so different, the basis of the value should be described in the scientific paper.
Thank you corrected.
6. Line 380: Equation (2); Table 1: Conceˆtemp_ice is not understandable quantity. The total volume of frazil ice can be calculated by integration over the water column and this value can be represented by thickness of ice. This quantity is easier to under- stand. Heat loss occurs at the ocean surface, and thus a quantity per unit area is more meaningful than a quantity per unit volume. I know that ice production represented by thickness per day is introduced using Conceˆtemp_ice later in section 6. But integrated frazil ice thickness should be introduced at this stage. This comment is also applied to Conceˆsalt_ice (Equation 5).

We had an error in our calculation for the column integral of ice production. This has changed our estimate of the total column integrals by about 10%. One of the authors felt that the standard and most intuitive way to present frazil ice inventories is to present them as a concentration in kg/m$^3$. We have followed this protocol, and the calculation of ice concentration is defined in section 4.1. Arguably, few people have intuition for frazil amounts, but the representation as concentration can be related to other quantities.

7. Line 398: How did you determine the starting location from below the anomaly?

There were a small number of profiles, so this procedure was done graphically, using the profiles as they are shown in Figure 7. We have revised Section 4.1 to clarify the approach.

"Because we lacked multiple profiles at the same location, we were not able to observe the time evolution of these anomalies.  Consequently, $T_b$ represents our best inference of the temperature of the water column prior to the onset of ice formation; it is highlighted in Figure 7a with the dashed line. We established the value of $T_b$ by averaging the temperature over a 10 m interval directly beneath the anomaly.  In most cases, this interval was nearly isothermal and isohaline, as would be expected within a well-mixed layer.  The uncertainty in the value of $T_b$ was estimated from the standard deviation within this 10 m interval; the average was $7.5 \times 10^{-5}$ °C, which is 1% of the temperature."

8. Line 417: Remove one of the double heat.
Thank you corrected.
9. Line 489: How "t" is finally represented using the known quantities?
Thank you, added from supplemental to main text.
10. Table 2: How "the life time" is finally represented (by an equation)?
Thank you, added.
11. Figure 10: color at Stn.32 looks like purple, not red.
Thank you figure updated.
12. Regarding the estimate of average or annual (monthly) ice production, there have been several satellite (microwave) investigations for these polynyas (e.g., Comiso et al., 2011; Drucker et al., 2011; Nihashi and Ohshima, 2015; Tamura et al., 2016), because the satellite microwave can provide daily sea ice condition. For example, data set of monthly ice production from Nihashi and Ohshima (2015) is now in public, and can be downloaded from http://www.lowtem.hokudai.ac.jp/wwwod/polar-seaflux/. As well as comparison with the model studies as was done in section 6.2, comparison and discussion with these satellite studies would enhance the value of this paper.

Thank you for pointing out these additional resources.  We have included these references in the manuscriptWe have revised section 6.2 – the discussion of previous sea ice production estimates. That section includes this paragraph on remote sensing: "Overall, these ice production estimates from in-situ data are larger than the seasonal production estimates derived from remote sensing products. Drucker et al (2011) used the AMSR-E instrument to obtain a seasonal average of 12 cm day$^{-1}$ for years 2003-2008. Oshima et al, (2016) estimated 6 cm day$^{-1}$ of seasonal production for the years 2003-2011, and Nihashi and Ohshima (2015) determined 7 cm day$^{-1}$ for years 2003-2010. Finally, Tamura et al (2016) found production rates that ranged from 7-13 cm day$^{-1}$, using both ECMWF and NCEP Reanalysis products for 1992-2013, reflecting a greater degree of consistency in successive estimates, likely because of consistency in the estimation methods. "

References
Comiso, J. C., R. Kwok, S. Martin, and A. L. Gordon, 2011: Variability and trends in sea ice extent and ice production in the Ross Sea. J. Geophys. Res., 116, C04021, doi:10.1029/2010JC006391.

[revised manuscript text omitted]

Commented [LDP5]: Correctedto reflect RC1- Comment 4. Given that we have the supplemental, I felt like i made sense to drop an equation.

[revised manuscript text omitted]

Commented [LDP10]: Sta 38 is not one of our stations

Commented [BL11]: This should probably go as a small figure in the manuscript

[Figure]

Empirical Relationship between Sensible Heat Flux and Sea Ice Production y = 0.1785x - 28.048
R² = 0.915

Figure 11: Empirical relationship between sensible heat flux and sea ice production: Production
rate = 0.1785 $Q_s$ -28.048, $R^2$ of 0.915.

Commented [BL12]: This doesn't need to be reported in the text, but can be shown in the Figure and should be reported in the figure caption.

Commented [BL13]: This is already in the ms

The met data from the NB Palmer and from Station Manuela (Figure 3) reveal that TNBP
experiences slower wind speeds and warmer temperatures than Station Manuela.  This
phenomenon has been explained as a consequence of adiabatic warming and a reduction in the
topographic 'Bernoulli' effects that cause wind speed to increase at Station Manuela (Schick,

Commented [LDP14]: Figure 3 n MS

2018). Before relating the time series of $Q_s$ from Manuela to the values of $Q_s$ computed for each CTD Station, we needed to account for the offset. The air temperatures were 6.5 °C warmer, and wind speed was on 7.5 m s$^{-1}$ slower in TNB, during the 13 days that the vessel was in the polynya, and these average differences were removed from the time series of $Q_s$ from Manuela. Figure S6.1 shows the corrected data against the original data.

We estimated the seasonal average in $Q_s$ over TNBP using the corrected met data from Station Manuela, and an average sea surface temperature from the CTD stations (-1.91 °C), the air density, specific heat capacity, and heat transfer coefficient remained the same as above. The average in $Q_s$ from April to September is 321 W m$^{-2}$. Using the empirical relationship described in Figure 11, the seasonal average ice production rate is 29 cm day$^{-1}$.

The seasonal sea ice production rate varies based on many factors affecting the rate of heat loss from the surface ocean. These factors include a strong negative feedback between ocean heat loss and sea ice cover. As the polynya builds up with ice, heat fluxes to the atmosphere will decline (Ackley et al, 2020 in review) until that ice cover is swept out of the polynya by the next katabatic wind event. This spatial variation in ice cover and wind speed, produces strong spatial gradients in the heat loss to the atmosphere that drives ice production. For example, Ackley et al., (Figure 3, 2020 in review) observed heat flux variations from nearly 2000 W m$^{-2}$ to less than 100 W m$^{-2}$ over less than 1 km. An integrated estimate of total polynya sea ice production should take these spatial gradients and the changes in polynya area into account. That analysis is somewhat beyond the scope of this study, but we anticipate including these ice production estimates within forthcoming sea ice production estimates for 2017 and PIPERS.

One interesting outcome of the scaling relationship in Figure 11, is the value of the y-intercept at 157 W m$^{-2}$. This relationship suggests that frazil ice production ceases when the heat flux falls below this range. This lower bound, in combination with the spatial gradients in heat flux may help to establish the region where active production is occurring.

**6.1 Comparison to prior model and field estimates of ice production**

The 29 cm d$^{-1}$ of seasonal average ice production that we estimated here, falls within the range of other in-situ ice production estimates. Schick (2018) estimated a seasonal average ice production rate of 15 cm day$^{-1}$, and Kurtz and Bromwich (1985), determined 30 cm day$^{-1}$. Both studies derived their ice production rates using a heat budget.

Overall, these ice production estimates from in-situ data are larger than the seasonal production estimates derived from remote sensing products. Drucker et al (2011) used the AMSR-E instrument to obtain a seasonal average of 12 cm day$^{-1}$ for years 2003-2008. Oshima et al, (2016) estimated 6 cm day$^{-1}$ of seasonal production for the years 2003-2011, and Nihashi and Ohshima (2015) determined 7 cm day$^{-1}$ for years 2003-2010. Finally, Tamura et al (2016) found production rates that ranged from 7-13 cm day$^{-1}$, using both ECMWF and NCEP Reanalysis products, reflecting a greater degree of consistency in successive estimates, likely because of consistency in the etimation methods.

In comparison, the modeling studies tend to skew higher than the in-situ observations: Sansiviero et al (2017) estimated 48 cm day$^{-1}$ using a sea-ice model. Petrelli, Bindoff, &

Bergamasco (2008) modeled a wintertime maximum production rates of 26.4 cm day$^{-1}$ using a coupled atmospheric-sea ice model. Fusco et al (2002) applied a model for latent heat polynyas and modeled production rate at 85 cm day$^{-1}$ for 1993 and 72 cm day$^{-1}$ for 1994.

It is worth noting that our production estimate applies to only frazil ice, rather than total
ice production. Columnar ice growth, for example, is usually considered the predominant ice
type in overall ice production, suggesting that our method implies a larger value for total ice
production during one season. However, the large range of ice production estimates cited above,
and the clustering of estimates from (1) in-situ data, from (2) remote sensing data, and from (3)
models in ranges that do not overlap, suggests a comparison of methods may be helpful to
achieve consistency and a convergence in estimates.

* * *
**Commented [BL19]:** Can you confirm we're comparing apples to apples? We want the average, if they provided it.

**Commented [BL20]:** Are these maxima or seasonal avgs?

**Commented [BL21]:** This paragraph should be revised if the modeling values you cite are not seasonal averages.

The spatial trend we observed somewhat mimics the model 3D model of TNBP from Gallee (1997) . During a four-day simulation, Gallee found highest ice production rates near the coast (e.g. our Station 35) of 50 cm day$^{-1}$, and decreasing production to 0 cm day$^{-1}$ downstream and at the outer boundaries, further west than PIPERS Station 33 (Figure 10). While some of the ice production rates derived from PIPERS CTD profiles exceed prior results, we attribute that excess to the relatively short time scale of these ice production "snapshots". These estimates integrate over minutes to tens of minutes, instead of days to months, therefore they are more likely to capture the high frequency variability in this ephemeral process. As the katabatic winds oscillate, the polynyas enter periods of slower ice production, driving average rates down. ¶
Our median production rate can be scaled over the average size of the polynya (1300 km2) and a season of March to October to find an annual production rate. The annual production rate is 76 km3 per year. In review of Drucker et al, 2011, they used Advanced Microwave Scanning Radiometer-EOS (AMSR-E) to find the average production rate of 88 km3. Oshima et al, (2016) used satellite remote sensing using passive microwave sensors to find the average production rate of 53 km3. Nihashi and Ohshima (2015) used passive microwave sensing to find the average production rate of 59 km3.¶

[revised manuscript text omitted]

Formatted Table

| Page 31: [14] Formatted | Lisa De Pace | 4/29/20 8:20:00 PM |
|---|---|---|

Font: Italic

| Page 35: [15] Deleted | Lisa De Pace | 4/30/20 3:06:00 PM |
|---|---|---|

| Page 35: [15] Deleted | Lisa De Pace | 4/30/20 3:06:00 PM |
|---|---|---|

| Page 35: [15] Deleted | Lisa De Pace | 4/30/20 3:06:00 PM |
|---|---|---|

| Page 35: [15] Deleted | Lisa De Pace | 4/30/20 3:06:00 PM |

---

## Author Response (AR2)

**Frazil ice growth and production during katabatic wind events in the Ross Sea, Antarctica**

| 1       | Lisa Thompson 1,5 , Madison Smith 2 , Jim Thomson 2 , Sharon Stammerjohn 3 , Steve Ackley 4 , and | Deleted: De Pace                                                                                 |
|---------|----------------------------------------------------------------------------------------------------------------------------------------------------------|--------------------------------------------------------------------------------------------------|
| 2       | Brice Loose 5                                                                                                                                 |                                                                                                  |
| 3       |                                                                                                                                                          |                                                                                                  |
| 4       | 1 Department of Science, US Coast Guard Academy, New London CT                                                                                |                                                                                                  |
| 5       | 2 Applied Physics Laboratory, University of Washington, Seattle WA                                                                            |                                                                                                  |
| 6       | 3 Institute for Arctic and Alpine Research. University of Colorado at Boulder, Boulder CO                                                     |                                                                                                  |
| 7       | 4 University of Teyes at San Antonio San Antonio TV                                                                                           | Deleted:                                                                                         |
| ,       |                                                                                                                                                          | During katabatic wind events                                                                     |
| 8       | Graduate School of Oceanography, University of Rhode Island, Narragansett RI                                                                             | Deleted: as deep as                                                                              |
| 9
10 | Correspondence to: Brice Loose (bloose@uri.edu)                                                                                                          | Deleted: Yet, upper ocean ter perfectly homogeneous, as we convective heat loss. Instead, |
| 11      |                                                                                                                                                          | Deleted: he                                                                                      |
| 112     | ABSTRACT: Katabatic wind in coastal polynyas expose the ocean to extreme heat loss causing                                                               | Deleted: and                                                                                     |
| 12      | interes assiss meduation and dance water formation around assets! Anteretion throughout                                                                  | Deleted: .                                                                                       |
| 13      | autumn and winter. Advancing sea ice and the extreme conditions, restrict direct observations of                                                         | Deleted: Considering both the water below, we suggest the is salinity reflects            |
| 15      | katabatic wind events in polynyas, impeding new insights into the evolution of these ice factories                                                       | Deleted: within the upper wa                                                                     |
| 16      | through the dark austral months. Here, we describe oceanic observations during multiple                                                                  | Deleted: We use                                                                                  |
| 17      | katehatia wind avanta in May. 2017 in the Tarre Nava Pay and Page See nalwayee, where wind                                                               | Deleted: a                                                                                       |
| 1/      | katabatic wind events in May, 2017 in the Terra Nova Bay and Ross Sea polynyas, where wind                                                               | Deleted: ified                                                                                   |
| 18      | speeds exceeded 20 m s -1 , air temperatures were below -25 °C, and the mixed layer extended to                                               | Deleted: to analyze                                                                              |
| 19      | 600 meters. Water column CTD, profiles revealed bulges of warm, salty water directly beneath                                                             | Deleted: to estimate                                                                             |
| 20      | the ocean surface and extending downwards tens of meters. These profiles suggest latent heat                                                             | Deleted: x 10 -3 and 13                                                               |
| 21      | and salt release during unconsolidated frazil ice production by atmospheric heat loss, a process                                                         | Deleted: by turbulent kinetic                                                                    |
| 22      | that has rarely if ever been observed outside the laboratory. A simple salt budget suggests these                                                        | Deleted: 7                                                                                       |
| 22      | anomalies reflect in situ frazil ice concentration that range over from 13 to 266 x $10^{-3}$ kg m -3                                         | Deleted: 12                                                                                      |
| 23      | Contemporaneous estimates of vertical mixing reveal rapid convection in these unstable density                                                           | Moved down [1]: The corresp
rates covary with wind speed
upstream-downstream length        |
| 25      | profiles, and mixing lifetimes from 12 to 7 minutes, respectively. The individual estimates of ice                                                       | Deleted: but they                                                                                |
| 26      | production from the salt budget reveal the intensity of short-term ice production, up to 110 cm d -                                           | Deleted: to a                                                                                    |
| 27      | 1 during the windiest events, and scaled-up seasonal average of 29 cm d -1 . We further found that                                 | Moved (insertion) [1]                                                                            |
| 20      | frequilies production rotes covery with wind speed and with lossifien along the unstream                                                                 | Deleted: The                                                                                     |
| 28      | mazin the production rates covary with while speed and with focation along the upstream-                                                                 | Deleted: corresponding                                                                           |
| 29      | downstream length of the polynya. These measurements reveal that it is possible to indirectly                                                            | Deleted:                                                                                         |
|         |                                                                                                                                                          |                                                                                                  |

**et, upper ocean temperature and salinity were not omogeneous, as would be expected with vigorous heat loss. Instead, t nd Considering both the colder air above and colder w, we suggest the increase in temperature and lects vithin the upper water column Ve use ïed analyze estimate etween $10^{\mbox{-}3}$ and 13y turbulent kinetic energy dissipation **wn [1]:** The corresponding frazil ice production ry with wind speed and with location along the downstream length of the polynya.**

[revised manuscript text omitted]